

# (Non) equilibrium dynamics: a (broken) symmetry of the Keldysh generating functional

**Camille Aron[1,2]⋆, Giulio Biroli[3,4] and Leticia F. Cugliandolo[5]**

**1** Laboratoire de Physique Théorique, École Normale Supérieure, CNRS,
PSL University, Sorbonne Université, 75005 Paris, France
**2** Instituut voor Theoretische Fysica, KU Leuven, Belgium
**3** Institut de Physique Théorique, CEA Saclay,
CNRS, 91191 Gif-sur-Yvette, France
**4** Laboratoire de Physique Statistique, École Normale Supérieure, CNRS,
PSL University, Sorbonne Université, 75005 Paris, France
**5** Laboratoire de Physique Théorique et Hautes Énergies,
Université Pierre et Marie Curie – Paris 6, Sorbonne Université, 75005 Paris, France

⋆ aron@lpt.ens.fr

## Abstract

We unveil the universal (model-independent) symmetry satisfied by Schwinger-Keldysh quantum field theories whenever they describe equilibrium dynamics. This is made possible by a generalization of the Schwinger-Keldysh path-integral formalism in which the physical time can be re-parametrized to arbitrary contours in the complex plane. Strong relations between correlation functions, such as the fluctuation-dissipation theorems, are derived as immediate consequences of this symmetry of equilibrium. In this view, quantum non-equilibrium dynamics – *e.g.* when driving with a time-dependent potential – are seen as symmetry-breaking processes. The symmetry-breaking terms of the action are identified as a measure of irreversibility, or entropy creation, defined at the level of a single quantum trajectory. Moreover, they are shown to obey quantum fluctuation theorems. These results extend stochastic thermodynamics to the quantum realm.

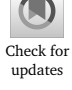

# 1   Introduction

## 1.1   Motivations

Encoded in the equilibrium dynamics of any system, there is a universal symmetry which entails fundamental consequences on equilibrium and out-of-equilibrium physics.

This strong assertion was demonstrated in the context of all *classical* systems. General properties of equilibrium dynamics, such as fluctuation-dissipation or Onsager reciprocal relations follow directly from this symmetry. For systems out of equilibrium, this symmetry is broken and this is intimately related to entropy production, as perhaps best epitomized by the Gallavotti-Cohen fluctuation theorems [1, 2].

More often than not, symmetries are easier to identify in a field-theory language as an invariance of an action, or more generally at the level of a generating functional. In the framework of the Martin-Siggia-Rose-Janssen-deDominicis (MSRJD) path-integral formulation of classical stochastic dynamics [3–5], the idea of an equilibrium symmetry of the action was first introduced by Janssen in the late 70s [6–8]. Recently, the authors showed that the (zero-source) MSRJD generating functional of equilibrium Langevin processes is invariant under a model-independent discrete field transformation combining time reversal with a thermal translation in the complex plane [9–11]. The above-mentioned consequences of this equilibrium symmetry (and its out-of-equilibrium breaking) were derived within this path-integral formalism, and they were generalized with little cost to virtually any type of classical equilib-

rium dynamics: from simple deterministic Newtonian dynamics, or basic Markov processes, to complex stochastic processes with colored and multiplicative noise.

Since then, the goal has been to generalize this fundamental symmetry to the case of *quantum* equilibrium dynamics, especially given the recent upsurge of activity on out-of-equilibrium quantum systems [12]. This is straightforward for those many-body systems the dynamics of which can be effectively described by classical stochastic processes (of quantum-mechanical origin) [13–15]. Related local symmetries have been proposed in the context of quantum hydrodynamics, *i.e.* for quantum systems that can be described by effective field theories for a few slow modes in local equilibrium with their environment, once the fast modes have been integrated out [16, 17]. However, the generalization to fully quantum-mechanical dynamics is an arduous challenge. In Ref. [18] this problem was solved in a setup in which the system is prepared in the far past and initial conditions can be neglected. This important progress provided a first identification of the fundamental symmetry encoded in quantum equilibrium dynamics. However, a full characterization of the symmetry taking into account the initial condition was still missing. This problem, whose solution is the main result of our work, was the main obstacle to address entropy production and quantum fluctuation theorems for which one needs to start from a given initial condition, drive the system out of equilibrium and break the symmetry.

In the last decades, classical thermodynamics was extended to systems with only a few degrees of freedom, for which fluctuations matter. One of the early successes in the field of stochastic thermodynamics has been to generalize usual thermodynamics quantities such as work, heat, or entropy creation, to single classical trajectories. Another resounding success is the discovery of the fluctuation theorems [1, 2]. The extension of the latter to the quantum realm was addressed starting from the early 2000s [19–21] and important progress has been achieved [22–24]. However, the thermodynamics of small *quantum* systems is still a subject of intense research. For instance, the notion of a continuous measurement and the precise definition of work performed along a single quantum trajectory are questions which are still debated. Experimentally, studying those topics is hard given that very few setups offer sufficient precision in the control of the system and its measurements. In this regard, a promising architecture is circuit quantum-electrodynamics with which the quantum trajectories of transmon qubits have already been monitored, and the first law of thermodynamics was verified along individual trajectories [25]. Our theoretical approach by-passes (the subtle) issues concerning the correct definition of work and entropy creation. By focusing on time-reversal symmetry and its breaking, it directly yields general relations—free of any presupposed definition—and important results on the entropy creation in quantum-mechanical systems.

The primary aim of our work is to generalize the classical approach of Ref. [9] and unveil the quantum version of the symmetry that encodes equilibrium dynamics in complete generality. Out of equilibrium, the symmetry is broken and we use this mechanism to derive the corresponding quantum fluctuation theorems. Incidentally, we solve a longstanding problem: the measure of the irreversibility generated along a single quantum trajectory is identified unambiguously. Unlike most previous results in the literature, the expressions obtained within our broken-symmetry approach are quite similar, in content and form, to the well-known classical results, thus shining a new light on modern quantum thermodynamics.

The derivation of our fundamental results relies on a prior overhaul of the conventional Schwinger-Keldysh path-integral formalism of quantum non-equilibrium dynamics. The main novelty brought by our approach is the possibility to re-parametrize the physical time with times that take values along generic contours in the complex plane. Developing a consistent way to change the time contour within a path integral is an old issue that already emerged in several field-theoretical problems [26, 27] and that we resolve here.

## 1.2 Physical framework

Concretely, we consider a generic quantum-mechanical system prepared, at the initial time $t = -t_0$, in an equilibrium state at temperature $\beta^{-1}$ with respect to the Hamiltonian[1] $H(-t_0)$. The subsequent unitary evolution is governed by the Hamiltonian $H(t)$, the time-dependence of which is assumed to be continuous and differentiable. Note that this does not exclude the case of a quantum quench as long as it can be seen as the limit of a continuous protocol performed on a time scale much smaller than any other time scale involved.

To present our results in the clearest manner, we simplify the discussion by considering the case of a system described by a non-relativistic real scalar field $\psi(t)$. We deliberately drop any spatial dependence of the field as it does not play any particular role in the derivation, and we choose to work within a symmetric time interval $t \in [-t_0, t_0]$. We also assume that the initial state is prepared within the canonical ensemble. However, for the sake of completeness, we display the bosonic (respectively fermionic) version of our main results for systems described by a complex scalar (respectively Grassmann) field. We also briefly discuss the case of a grand-canonical initial state. Moreover, we assume that the system is neither subjected to magnetic fields nor coupled to any quantity that is odd under time-reversal: $\Theta H(t)\Theta = H(t)$ where $\Theta$ is the antilinear time-reversal operator. The generalizations to time intervals that are not symmetric, fields that are fully space- and time-dependent, or vectors fields, are straightforward. Unless otherwise stated we set $k_{\mathrm{B}} = \hbar = 1$ throughout the manuscript. In Sect. 5 we restore $\hbar$ to take the classical limit.

## 1.3 Main results

Let us outline the key steps followed in the presentation and point to the main results presented in this manuscript.

**Formalism.** In Sect. 2, we start from the operator formulation of the Kadanoff-Baym generating functional $Z[J^+, J^-]$ and we modify it to construct a path-integral representation where time can now run on a generic contour in the complex plane rather than on the real line. This flexibility of the formalism is crucial to identify the symmetry of equilibrium dynamics. The final expression for $Z[J^+, J^-]$ is given in Eq. (26). Compared to the conventional Kadanoff-Baym construction, the most important step is the generalization of the resolution of the identity which is usually inserted at the time slice $t$ to derive the path integral from the operator formalism:

$$\mathbb{I} = \int \mathrm{d}\psi(t) \, |\psi(t)\rangle\langle\psi(t)| \quad \text{to} \quad \mathbb{I} = \int \mathrm{d}\psi(t) \, \mathrm{e}^{+\mathrm{i}\theta(t)H(t)}|\psi(t)\rangle\langle\psi(t)|\mathrm{e}^{-\mathrm{i}\theta(t)H(t)}, \quad (1)$$

where $\theta(t)$ is a complex function that appears in the re-parametrization of time as $\tau(t) \equiv t + \theta(t)$.

Note that within a real-time path integral over a field $\psi(t)$ with $t \in [-t_0, t_0]$, it is important to realize that expressions of the type $\psi(-t + \mathrm{i}\beta/2)$ have *a priori* no meaning, and that they *cannot* be given one by an infinite Taylor expansion $\psi(-t + \mathrm{i}\beta/2) \equiv \psi(-t) - \mathrm{i}\beta/2\partial_t\psi(-t) - (\beta/2)^2/2! \, \partial_t^2\psi(-t) + \dots$ because $(i)$ this would require $\psi(t)$ to be analytic at all times, whereas the functional integral is performed over *all* fields, continuous or not, differentiable or not; $(ii)$ the left-hand side of the above expansion is real whereas its right-hand side is not for most trajectories of $\psi(t)$. Having said that, this proviso can be relaxed inside the source term of a generating functional. Indeed, since sources are set to zero at the end of any physical computation, the fields to which they couple

---

[1]In practice, for any initial density matrix $\rho(-t_0)$, it is always possible to find a Hermitian operator $H(-t_0)$ such as $\rho(-t_0) = \exp[-\beta H(-t_0)]/\mathscr{Z}(-t_0)$.

only appear at the level of expectation values, and can therefore be attributed the regularity properties of typical quantum-mechanical observables. See the discussion below Eq. (34). In Ref. [18], these issues were solved by working in the frequency domain, where the expression $\psi(-t + i\beta/2)$ corresponds to $e^{\beta\omega/2}\psi(\omega)^*$. However, working with Fourier modes is possible only for equilibrium systems that have been prepared in the far past, $-t_0 \mapsto -\infty$. In order to obtain an expression of the symmetry valid in the time domain and with initial conditions at a finite time—an essential ingredient to discuss its breaking in out of equilibrium settings—one has to give a direct meaning to $\psi(-t + i\beta/2)$, which we do through a re-parametrization of time allowing the deformation of the time contour in the complex plane.

**Equilibrium dynamics.** In Sect. 3, we specialize the discussion to the case of equilibrium dynamics. We propose a time contour $\mathscr{C}_\beta$, represented in Fig. 3, that is suitable for identifying the equilibrium symmetry. The corresponding generating functional $Z_\beta[J_\beta^+, J_\beta^-]$ is given in Eq. (43), and the equilibrium symmetry is presented around Eq. (46). The fluctuation-dissipation theorem is derived as a corresponding Ward-Takahashi identity in Eq. (69).

**Out-of-equilibrium dynamics.** In Sect. 4, we break the symmetry by driving the system out of equilibrium with a time-dependent protocol controlled externally *via* the parameter $\lambda(t)$. We exploit the symmetry-breaking terms of the action to derive the quantum Jarzynski equality in Eq. (101) and the quantum Crooks fluctuation theorem in Eq. (104). We also generalize the approach to derive novel relations for multi-time correlation functions. As a by-product, we identify in Eq. (92), the general expression of the entropy production along a single trajectory of a quantum driven system, $\Sigma$. It involves the operator $\dot{\sigma}(t) = -2 e^{\beta H(t)/4} \partial_t \left[ e^{-\beta H(t)/2} \right] e^{\beta H(t)/4}$ which quantifies the rate of production of irreversibility. The corresponding measure of irreversibility generated along a single trajectory, $\mathscr{S}^{\text{irr}}$, is given in Eq. (106). The physical results are presented in such a way to be accessible to the reader unfamiliar with the Keldysh formalism. To illustrate our findings, we present the case of a two-level system driven out of equilibrium by a time-dependent Zeeman field. The dynamics are solved numerically. This simple quantum-mechanical example can be relatively simply implemented within a superconducting-qubit architecture, where the two-level system may be monitored and manipulated *via* its coupling to a photonic cavity mode, see for instance Ref. [25].

**Classical dynamics.** In Sect. 5, we take the classical limit of our formalism and we recover, after a mere redefinition of the response field $i\hat{\psi}(t) \to i\hat{\psi}(t) - \beta \partial_t \psi(t)$, the Martin-Siggia-Rose-Janssen-deDominicis formalism and the classical results of Ref. [9].

**Conclusions.** Finally, we present our conclusions and an outlook in Sect. 6.

## 2 Deformation of the Kadanoff-Baym time contour

In this Section, we first briefly recall the Keldysh path-integral formalism, in particular the Kadanoff-Baym construction where fields are integrated over an initial Matsubara-like imaginary-time branch followed by a close contour in real time. As we shall see, there is a degree of arbitrariness in the choice of the Kadanoff-Baym time contour. For example, one often finds in the literature that the Matsubara-like branch follows the real-time evolution rather than preceding it. Those are of course equivalent formulations. In this Section, we generalize

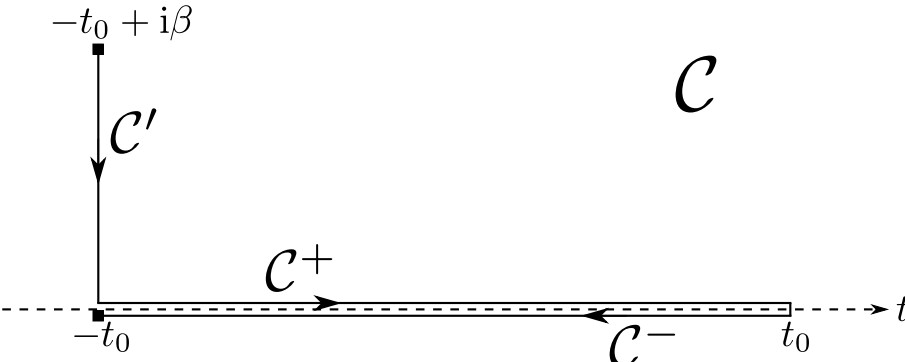

Figure 1: Kadanoff-Baym contour $\mathscr{C}$ used in the generating functional in Eq. (2). It has an imaginary branch $\mathscr{C}'$ and two real-time branches, $\mathscr{C}^+$ and $\mathscr{C}^-$. Arrows indicate the time ordering $\mathrm{T}_{\mathscr{C}}$.

and formalize this flexibility by constructing a path-integral representation of the generating functional which allows us to define and integrate the fields over *generic complex-time contours* rather than over the conventional Kadanoff-Baym contour. The reader eager to have a concise illustration of this idea may consult the Appendix A where we discuss it in the context of the operator formalism.

## 2.1 Recap of the Keldysh formalism

Let us start with a quick reminder of the derivation of the conventional path-integral formulation of the Keldysh generating functional [28,29]. We recommend Ref. [30] for a detailed and pedagogical presentation of this technique. In the operator formalism, the so-called Kadanoff-Baym generating functional reads

$$Z[J^+, J^-] = \mathrm{Tr}\left[\mathrm{T}_{\mathscr{C}} \, \mathrm{e}^{-\mathrm{i}\int_{\mathscr{C}} \mathrm{d}t \, H(t)} \, \mathrm{e}^{\int_{-t_0}^{t_0} \mathrm{d}t \, J^+(t)\psi - J^-(t)\psi}\right]\Big/ \mathscr{Z}(-t_0), \tag{2}$$

where $\mathrm{T}_{\mathscr{C}}$ is the time-ordering operator along the Kadanoff-Baym contour $\mathscr{C}$ represented in Fig. 1. It has an initial Matsubara-like imaginary-time branch $\mathscr{C}'$ from $-t_0 + \mathrm{i}\beta$ to $-t_0$ that generates the initial density matrix $\mathrm{e}^{-\beta H(-t_0)}/\mathscr{Z}(-t_0)$, a forward real-time branch $\mathscr{C}^+$ from $-t_0$ to $t_0$, and a backward real-time branch $\mathscr{C}^-$ from $t_0$ back to $-t_0$. The trace enforces equal boundary conditions at $-t_0 + \mathrm{i}\beta$ and $-t_0$. Altogether, $\mathscr{C} \equiv \mathscr{C}' \cup \mathscr{C}^+ \cup \mathscr{C}^-$. Objects defined on the forward and backward branches are indicated by the superscripts $+$ and $-$, respectively. The sources $J^+(t)$ and $J^-(t)$ are coupled to the field $\psi$ on the forward and backward branches, respectively. The initial partition function $\mathscr{Z}(-t_0) \equiv \mathrm{Tr}\,\mathrm{e}^{-\beta H(-t_0)}$ ensures that the zero-source generating functional is system-independent, $Z[J^+ = J^- = 0] = 1$.

In order to obtain the path-integral representation of $Z[J^+, J^-]$, one slices the real-time branches $\mathscr{C}^+$ and $\mathscr{C}^-$ in infinitesimal time steps $\mathrm{d}t$, inserts the following resolution of the identity at each time step

$$\mathbb{I} = \int \mathrm{d}\psi(t) \, |\psi(t)\rangle\langle\psi(t)|, \tag{3}$$

and uses the usual Feynman construction

$$\langle\psi(t + \mathrm{d}t)|\mathrm{e}^{-\mathrm{i}H(t)\mathrm{d}t}|\psi(t)\rangle \sim \mathrm{e}^{\mathrm{i}\mathscr{L}[\psi(t);t]\mathrm{d}t}, \tag{4}$$

where $\mathscr{L}$ is the Lagrangian corresponding to the Hamiltonian $H$. In this way, one ends up with the following well-known path-integral representation of the Kadanoff-Baym generating functional:

$$Z[J^+, J^-] = \mathscr{Z}(-t_0)^{-1} \int \mathscr{D}[\psi^+, \psi^-] \langle \psi^+(-t_0)|e^{-\beta H(-t_0)}|\psi^-(-t_0)\rangle \langle \psi^-(t_0)|\psi^+(t_0)\rangle$$

$$\times \exp\left( i \int_{-t_0}^{t_0} dt \, \mathscr{L}[\psi^+(t); t] - \mathscr{L}[\psi^-(t); t] \right)$$

$$\times \exp\left( \int_{-t_0}^{t_0} dt \, J^+(t)\psi^+(t) - J^-(t)\psi^-(t) \right). \tag{5}$$

In the above procedure, the number of degrees of freedom was doubled: the functional integral over the field $\psi^+(t)$ corresponds to the time evolution on the forward branch $\mathscr{C}^+$, while the integral over $\psi^-(t)$ corresponds to the evolution on the backward branch $\mathscr{C}^-$. All irrelevant prefactors were included in the functional measure $\mathscr{D}[\psi^+, \psi^-]$. Notice the presence of the matrix element $\langle \psi^-(t_0)|\psi^+(t_0)\rangle$ in Eq. (5). It couples the two copies of the system at the final time $t_0$. It is bound to play a role as crucial as the one of the initial density matrix element $\langle \psi^+(-t_0)|e^{-\beta H(-t_0)}|\psi^-(-t_0)\rangle$.

Proceeding similarly, one can slice $e^{-\beta H(-t_0)}$ in infinitesimal imaginary time steps and introduce a path integral where the Lagrangian is integrated over the contour $\mathscr{C}'$ running from $-t_0 + i\beta$ to $-t_0$. Note that there is some degree of arbitrariness in the choice of $\mathscr{C}'$: the cyclic property of the trace in Eq. (2) also allows us to define it between $-t_0$ and $-t_0 - i\beta$. Below, we further discuss the different forms that the complete contour can take.

## 2.2 Time-dependent resolution of the identity

Let us start with the operator expression of the Kadanoff-Baym generating functional given in Eq. (2). Rather than using the resolution of the identity given in Eq. (3), let us transform it to an energy-dependent rotating frame and use the following time-dependent identity:

$$\mathbb{I} = \int d\psi^a(t) \, e^{+i\theta^a(t)H(t)}|\psi^a(t)\rangle\langle\psi^a(t)|e^{-i\theta^a(t)H(t)}, \tag{6}$$

where $a$ is either $a = +$ or $a = -$ and $\theta^a(t)$ is a generic complex function, continuous and differentiable on $t \in [-t_0, t_0]$. The identity in Eq. (6) is easily obtained by acting on both sides of Eq. (3) with $\exp[\pm i\theta^a(t)H(t)]$. Following similar steps as in the conventional Kadanoff-Baym construction, we now obtain

$$Z[J^+, J^-] = \mathscr{Z}(-t_0)^{-1} \int \mathscr{D}[\psi^+, \psi^-] \langle \psi^+(-t_0)|e^{-[i\theta^+(-t_0)-i\theta^-(-t_0)+\beta]H(-t_0)}|\psi^-(-t_0)\rangle$$

$$\times \langle \psi^-(t_0)|e^{i[\theta^+(t_0)-\theta^-(t_0)]H(t_0)}|\psi^+(t_0)\rangle$$

$$\times \exp\left( i \int_{-t_0}^{t_0} dt \, \mathscr{L}_\theta^+[\psi^+(t); t] - \mathscr{L}_\theta^-[\psi^-(t); t] \right)$$

$$\times \exp\left( \int_{-t_0}^{t_0} dt \, J^+(t)\psi_\theta^+(t) - J^-(t)\psi_\theta^-(t) \right), \tag{7}$$

with the initial partition function $\mathscr{Z}(-t_0) \equiv \mathrm{Tr}\, e^{-\beta H(-t_0)}$. The explicit functional forms of the Lagrangians $\mathscr{L}_\theta^a$ and of the fields coupling to the sources, $\psi_\theta^a(t)$, will be given below. Note that the $t = t_0$ matrix element of the original Kadanoff-Baym generating functional in Eq. (5) transformed into a term that is now quite similar, in nature, to the initial density matrix element at $t = -t_0$.

**Operator insertion.** In the conventional Kadanoff-Baym formalism, one computes the expectation value of an operator $X$ at time $t$ by inserting the operator $X_{\mathrm{H}}^a(t)$ within the trace of Eq. (2) or, equivalently, by inserting the matrix component $\langle \psi^a(t)|X|\psi^a(t)\rangle$ within the path integral summation of Eq. (5). Similar rules apply within our modified formalism, except that operators must now first be modified according to

$$X \mapsto X_\theta^a(t) \equiv \mathrm{e}^{-i\theta^a(t)H(t)} X\, \mathrm{e}^{i\theta^a(t)H(t)} = \mathrm{e}^{-i\theta^a(t)\mathrm{Ad}_{H(t)}} X\,, \tag{8}$$

with $\mathrm{Ad}_Y X \equiv [Y;X]$. In particular, in the source term of Eq. (7), we introduced

$$\psi_\theta^a(t) \equiv \langle \psi^a(t)|\mathrm{e}^{-i\theta^a(t)\mathrm{Ad}_{H(t)}}\psi|\psi^a(t)\rangle\,. \tag{9}$$

## 2.3 Unithermal Hamiltonian

The new Lagrangians $\mathscr{L}_\theta^+$ and $\mathscr{L}_\theta^-$ in Eq. (7) originate from the matrix elements that appear between two consecutive insertions of the resolution of the identity:

$$\langle \psi^+(t+\mathrm{d}t)|\mathrm{e}^{-i\theta^+(t+\mathrm{d}t)H(t+\mathrm{d}t)}\mathrm{e}^{-iH(t)\mathrm{d}t}\mathrm{e}^{i\theta^+(t)H(t)}|\psi^+(t)\rangle \sim \mathrm{e}^{i\mathscr{L}_\theta^+[\psi^+(t);t]\mathrm{d}t}\,, \tag{10}$$

$$\langle \psi^-(t)|\mathrm{e}^{-i\theta^-(t)H(t)}\mathrm{e}^{iH(t)\mathrm{d}t}\mathrm{e}^{i\theta^-(t+\mathrm{d}t)H(t+\mathrm{d}t)}|\psi^-(t+\mathrm{d}t)\rangle \sim \mathrm{e}^{-i\mathscr{L}_\theta^-[\psi^-(t);t]\mathrm{d}t}\,. \tag{11}$$

In order to identify $\mathscr{L}_\theta^a$ we first focus on its corresponding Hamiltonian $H_\theta^a$. This is obtained by expanding the exponentials in the left-hand side up to first order in $\mathrm{d}t$ and equating the result with $\mathbb{I} - a\,i\,\mathrm{d}t\,H_\theta^a$. For $H_\theta^+$ (and similarly for $H_\theta^-$), one has

$$\mathrm{e}^{-i\theta^+(t+\mathrm{d}t)H(t+\mathrm{d}t)}\,\mathrm{e}^{-iH(t)\mathrm{d}t}\,\mathrm{e}^{i\theta^+(t)H(t)}$$
$$= \mathrm{e}^{-i\theta^+(t)H(t)-\mathrm{d}t\,\partial_t[i\theta^+(t)H(t)]}\,\mathrm{e}^{i\theta^+(t)H(t)}\big[\mathbb{I}-i\,\mathrm{d}t\,H(t)\big] + \mathcal{O}(\mathrm{d}t^2) \tag{12}$$
$$= \mathbb{I}-i\,\mathrm{d}t\,H_\theta^+(t) + \mathcal{O}(\mathrm{d}t^2)\,, \tag{13}$$

where we identified the non-Hermitian Hamiltonian

$$H_\theta^a(t) \equiv H(t) + \int_0^1 \mathrm{d}x\,\mathrm{e}^{-xi\theta^a(t)\mathrm{Ad}_{H(t)}}\frac{\partial}{\partial t}\big(\theta^a(t)H(t)\big)\,. \tag{14}$$

Since the Hamiltonian $H_\theta^a(t)$ can be seen as the combination of a time-dependent unitary rotation and a thermal rotation of the original Hamiltonian $H(t)$, we call it "unithermal". Its explicit expression in power series of $\theta^a$ reads

$$H_\theta^a(t) = H(t) + \partial_t[\theta^a(t)H(t)] - (i/4)\big[\theta^a(t)H(t), \partial_t[\theta^a(t)H(t)]\big] + \dots \tag{15}$$

Note that each commutator of $H(t)$ with $\partial_t H(t)$ typically brings a factor of $\hbar$.

As we shall see later, it can prove useful to divide $H_\theta^a(t)$ into instantaneous and varying components, $\overline{H}_\theta^a(t)$ and $\widetilde{H}_\theta^a(t)$ respectively,

$$H_\theta^a(t) = \overline{H}_\theta^a(t) + \widetilde{H}_\theta^a(t)\,, \tag{16}$$

with

$$\overline{H}_\theta^a(t) = [1 + \partial_t\theta^a(t)]H(t)\,, \tag{17}$$

$$\widetilde{H}_\theta^a(t) = \theta^a(t)\int_0^1 \mathrm{d}x\,\mathrm{e}^{-xi\theta^a(t)\mathrm{Ad}_{H(t)}}\,\partial_t H(t)\,. \tag{18}$$

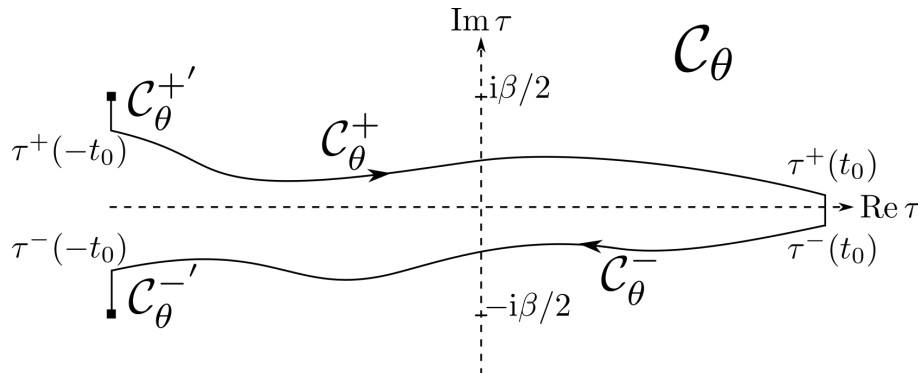

Figure 2: A possible complex-time contour, $\mathscr{C}_\theta$. The black squares mark the boundary condition $\psi^+(\tau^+(-t_0) + \mathrm{i}\beta/2) = \psi^-(\tau^+(-t_0) - \mathrm{i}\beta/2)$.

## 2.4 Complex time re-parametrization and Lagrangians

In order to obtain the corresponding unithermal Lagrangian, $\mathscr{L}_\theta^a = \overline{\mathscr{L}}_\theta^a + \widetilde{\mathscr{L}}_\theta^a$, we first perform the following time re-parametrization of the fields:

$$\psi^a(t) \mapsto \psi^a(\tau^a(t)), \tag{19}$$

with the complex map

$$\tau^a(t) \equiv t + \theta^a(t) \tag{20}$$

which is assumed to have a well-defined inverse $t^a(\tau)$. When $t$ runs over the contour $\mathscr{C}^+$ (respectively $\mathscr{C}^-$), $\tau$ runs over the complex-time contour $\mathscr{C}_\theta^+ \equiv \{\tau^+(t), t \in \mathscr{C}^+\}$ (respectively $\mathscr{C}_\theta^- \equiv \{\tau^-(t), t \in \mathscr{C}^-\}$). In this manuscript, we use Latin letters such as $t$, $t'$ to denote times on the original contour $\mathscr{C}$, while the Greek letters $\tau$, $\tau'$ are used for complex times running on the newly defined contour[2] $\mathscr{C}_\theta$, see Fig. 2. After this time re-parametrization, the expression of the varying component of the unithermal Lagrangian reads

$$\widetilde{\mathscr{L}}_\theta^a[\psi^a(\tau^a(t)); t] = -\theta^a(t) \int_0^1 \mathrm{d}x \, \langle \psi^a(\tau^a(t)) | \mathrm{e}^{-x\mathrm{i}\theta^a(t)\mathrm{Ad}_{H(t)}} \, \partial_t H(t) | \psi^a(\tau^a(t)) \rangle, \tag{21}$$

whereas the instantaneous component of the unithermal Lagrangian, $\overline{\mathscr{L}}_\theta^a$, simply boils down to the original Lagrangian $\mathscr{L}$:

$$\mathrm{d}t \, \overline{\mathscr{L}}_\theta^a[\psi^a(t + \theta^a(t)); t] = \mathrm{d}\tau \, \mathscr{L}[\psi^a(\tau); t^a(\tau)]. \tag{22}$$

This can easily be checked for a simple massive particle of coordinate $\psi$ and conjugate momentum $\pi$ in a time dependent potential $V(\psi; t)$. The Hamiltonian $H(t) = \frac{\pi^2}{2m} + V(\psi; t)$ yields $\overline{H}_\theta^a(t) = \partial_t \tau^a(t) H(t)$, and ultimately,

$$\overline{\mathscr{L}}_\theta^a[\psi(\tau^a(t)); t] = \frac{1}{2} m \frac{1}{\partial_t \tau^a(t)} [\partial_t \psi(\tau^a(t))]^2 - \partial_t \tau^a(t) V(\psi^a(\tau^a(t)); t) \tag{23}$$

$$= \partial_t \tau^a(t) \left( \frac{1}{2} m [\partial_{\tau^a} \psi(\tau^a)]^2 - V(\psi^a(\tau^a); t) \right) = \frac{\mathrm{d}\tau^a}{\mathrm{d}t} \mathscr{L}[\psi^a(\tau); t^a(\tau)].$$

[2]In our formalism, together with the deformation of the time contour, we also modify the operators that are to be probed on the contour: they include the necessary exponential factors, see Eq. (8), that ensure the overall convergence of all expressions. At any point, one can re-express the path-integrals in the language of the operator formalism, and come back to a simple Kadanoff-Baym expression of real-time correlators (without having to use any analyticity property). Therefore, the new time-ordered contour $\mathscr{C}_\theta$ is not restricted to having a downward slope at every point.

## 2.5 Generating functional

For simplicity, we restrict the analysis to functions $\theta^a(t)$ satisfying the boundary conditions[3] $\operatorname{Re}\theta^+(-t_0) = \operatorname{Re}\theta^-(-t_0)$ and $\operatorname{Re}\theta^+(t_0) = \operatorname{Re}\theta^-(t_0)$. We represent the $t = \pm t_0$ matrix elements in Eq. (7) by the following path integrals:

$$\langle\psi^+(\tau^+(-t_0))|e^{-[i\theta^+(-t_0)-i\theta^-(-t_0)+\beta]H(-t_0)}|\psi^-(\tau^-(-t_0))\rangle$$
$$= \int \mathscr{D}[\psi^+,\psi^-]\, e^{i\int_{\operatorname{Re}\tau^+(-t_0)+i\beta/2}^{\tau^+(-t_0)}d\tau\,\mathscr{L}[\psi^+(\tau);-t_0]+i\int_{\tau^-(-t_0)}^{\operatorname{Re}\tau^-(-t_0)-i\beta/2}d\tau\,\mathscr{L}[\psi^-(\tau);-t_0]}, \tag{24}$$

$$\langle\psi^-(\tau^-(t_0))|e^{i[\theta^+(t_0)-\theta^-(t_0)]H(t_0)}|\psi^+(\tau^+(t_0))\rangle$$
$$= \int \mathscr{D}[\psi^+,\psi^-]\, e^{i\int_{\tau^+(t_0)}^{\operatorname{Re}\tau^+(t_0)}d\tau\,\mathscr{L}[\psi^+(\tau);t_0]+i\int_{\operatorname{Re}\tau^-(t_0)}^{\tau^-(t_0)}d\tau\,\mathscr{L}[\psi^-(\tau);t_0]}, \tag{25}$$

together with the boundary conditions $\psi^+\big(\operatorname{Re}\tau^+(-t_0)+i\beta/2\big) = \psi^-\big(\operatorname{Re}\tau^-(-t_0)-i\beta/2\big)$ and $\psi^+\big(\operatorname{Re}\tau^+(t_0)\big) = \psi^-\big(\operatorname{Re}\tau^-(t_0)\big)$. Equations (24) and (25) correspond to integrating the original Lagrangian $\mathscr{L}$ along four new purely imaginary (vertical) branches that we gather in

$$\mathscr{C}_\theta^{+'} \equiv [\operatorname{Re}\tau^+(-t_0)+i\beta/2, \tau^+(-t_0)]\cup[\tau^+(t_0), \operatorname{Re}\tau^+(t_0)]$$

and

$$\mathscr{C}_\theta^{-'} \equiv [\operatorname{Re}\tau^-(t_0), \tau^-(t_0)]\cup[\tau^-(-t_0), \operatorname{Re}\tau^-(t_0)-i\beta/2].$$

Therefore, we can extend the complex-time contour $\mathscr{C}_\theta$ defined in Sect. 2.4 to $\mathscr{C}_\theta \equiv \mathscr{C}_\theta^+ \cup \mathscr{C}_\theta^{+'} \cup \mathscr{C}_\theta^- \cup \mathscr{C}_\theta^{-'}$, see Fig. 2, so as to express the generating functional in Eq. (7) in the form

$$Z[J^+,J^-] = \mathscr{Z}(-t_0)^{-1}\int \mathscr{D}[\psi^+,\psi^-]\, \exp\big(iS_\theta[\psi^+,\psi^-]\big)$$
$$\times \exp\int_{-t_0}^{t_0}dt\, \big(J^+(t)\psi_\theta^+(t)-J^-(t)\psi_\theta^-(t)\big), \tag{26}$$

with the action

$$S_\theta[\psi^+,\psi^-] = \int_{\mathscr{C}_\theta}d\tau\,\mathscr{L}[\psi(\tau);t(\tau)]$$
$$+\int_{-t_0}^{t_0}dt\, \big(\widetilde{\mathscr{L}}_\theta^+[\psi^+(\tau^+(t));t]-\widetilde{\mathscr{L}}_\theta^-[\psi^-(\tau^-(t));t]\big). \tag{27}$$

The expression above has three contributions: (*i*) the instantaneous component, originating from $\overline{H}_\theta^a(t)$, where $\mathscr{L}$ is the original Lagrangian (we omitted the branch superscripts); (*ii*) the varying component, originating from $\widetilde{H}_\theta^a(t)$, where $\widetilde{\mathscr{L}}_\theta^a$ is expressed in Eq. (21); and (*iii*) the source term where we redefined

$$\psi_\theta^a(t) \equiv \langle\psi^a(\tau^a(t))|e^{-i\theta^a(t)\operatorname{Ad}_{H(t)}}\psi|\psi^a(\tau^a(t))\rangle. \tag{28}$$

**Bosonic and fermionic versions.** In the case of a bosonic (respectively fermionic) path integral, where the field $\psi(t)$ is complex-valued (respectively Grassmann-valued), generating

---

[3]This guarantees that the matrix elements in Eqs. (24) and (25) can be represented by imaginary-time path integrals, *i.e.* the contours $\mathscr{C}_\theta^{+'}$ and $\mathscr{C}_\theta^{-'}$ are vertical in the complex plane.

functionals of similar form are obtained. The resolution of the identity analogous to the one in Eq. (6) uses the coherent-state representation, and reads

$$\mathbb{I} = \int \frac{\mathrm{d}\psi^a(t)^* \mathrm{d}\psi^a(t)}{\mathcal{N}} \, \mathrm{e}^{-\psi^a(t)^*\psi^a(t)} \, \mathrm{e}^{+\mathrm{i}\theta^a(t)H(t)} |\psi^a(t)\rangle \langle\psi^a(t)| \mathrm{e}^{-\mathrm{i}\theta^a(t)H(t)}, \qquad (29)$$

where $\psi^*$ denotes the complex (respectively Grassmann) partner and $\mathcal{N} = 2\pi\mathrm{i}$ (respectively $\mathcal{N} = 1$) for bosons (respectively fermions). The resulting Lagrangian reads

$$\mathcal{L}_\theta^a[\psi^a(t); t] = \psi^a(t)^* \mathrm{i}\partial_t \psi^a(t) - :H_\theta^a: \left( \psi^+(t)^*, \psi^+(t); t \right), \qquad (30)$$

where $:H_\theta^a:$ denotes the normal ordered form of $H_\theta^a$ defined in Eq. (15).

Note that when working with initial conditions in the grand-canonical rather than canonical ensemble, *i.e.* $\rho(-t_0) \sim \mathrm{e}^{-\beta(H(-t_0)-\mu N)}$ where $N$ is the operator counting the total number of particles (assumed to be conserved by the dynamics) and $\mu$ is the chemical potential, we need to replace $H(t)$ in the resolution of the identity above with $H(t) - \mu N$. While we leave the derivation of the corresponding generating functional as an exercise for the reader, we mention that it now involves the Lagrangians $\mathcal{L} \mapsto \mathcal{L} + \mu \psi^{a*}\psi^a$ on the Matsubara-like branches $\mathscr{C}_\theta^{+'}$ and $\mathscr{C}_\theta^{-'}$, and that its source term reads

$$\sum_{a=\pm} \int_{\mathscr{C}^a} \mathrm{d}t \left( J^a(t)^* \psi_\theta^a(t) + \psi_\theta^a(t)^* J^a(t) \right), \qquad (31)$$

with

$$\psi_\theta^a(t) \equiv \mathrm{e}^{-\mathrm{i}\theta^a(t)\mu} \langle\psi^a(\tau^a(t))| \, \mathrm{e}^{-\mathrm{i}\theta^a(t)\mathrm{Ad}_{H(t)}} \psi \, |\psi^a(\tau^a(t))\rangle, \qquad (32)$$

$$\psi_\theta^a(t)^* \equiv \mathrm{e}^{\mathrm{i}\theta^a(t)\mu} \langle\psi^a(\tau^a(t))| \, \mathrm{e}^{-\mathrm{i}\theta^a(t)\mathrm{Ad}_{H(t)}} \psi^\dagger \, |\psi^a(\tau^a(t))\rangle. \qquad (33)$$

## 3 The fundamental symmetry of equilibrium dynamics

In this Section, we restrict the analysis to the case of equilibrium dynamics. Equilibrium dynamics are guaranteed provided that the system is prepared in equilibrium, *i.e.* described by a Gibbs-Boltzmann thermal density matrix $\rho(-t_0) \sim \exp(-\beta H)$ in the canonical ensemble, or $\rho(-t_0) \sim \exp(-\beta(H - \mu N))$ in the grand-canonical ensemble, and time-evolved with the same time-independent Hamiltonian $H$. In case the system were not isolated, its environment must also remain in equilibrium at the same temperature $\beta^{-1}$ and chemical potential $\mu$.

### 3.1 Generating functional on a generic contour

For a constant Hamiltonian $H$ and a generic $\theta^a(t)$, the varying component of the unithermal Lagrangian in the generating functional $Z[J^+, J^-]$ expressed in Eq. (26) vanishes, $\widetilde{\mathcal{L}_\theta^a} = 0$. Moreover, the quantity $\psi_\theta^a(t)$, entering the source term and defined in Eq. (28), can be massaged by using the simple relation between the unithermal and the original Hamiltonians, $H_\theta^a(t) = [1 + \partial_t \theta^a(t)] H$, and it can be re-written in terms of the re-parametrized time $\tau$ as

$$\psi_\theta^a(\tau) = \langle\psi^a(\tau)| \mathrm{e}^{-\mathrm{i}\theta^a(\tau)\partial_\tau t^a \, \mathrm{Ad}_{H_\theta^a(t^a(\tau))}} \psi |\psi^a(\tau)\rangle, \qquad (34)$$

where we used the shorthand notation $\theta^a(\tau) \equiv \theta^a(t^a(\tau))$.

It is important to note that within a path integral, the summation is performed over *all* field trajectories, including configurations that are not continuous. Therefore, in principle, $\psi_\theta^a(\tau)$ cannot be analytic as the expression in Eq. (36) seems to suggest. Time derivatives

such as $\partial_\tau^n \psi_\theta^a(\tau)$ are ill-defined. However, given that the source term of the generating functional is set to zero at the end of any physical computation, $\psi_\theta^a(\tau)$ will only appear at the level of observables, *i.e.* not in the path measure but in averaged quantities of the type $\langle \psi_\theta^a(\tau) \psi_\theta^b(\tau') \ldots \rangle$ for which analyticity is restored and time derivatives can be properly defined *via* $\langle \partial_\tau^n \psi_\theta^a(\tau) \psi_\theta^b(\tau') \ldots \rangle = \partial_\tau^n \langle \psi_\theta^a(\tau) \psi_\theta^b(\tau') \ldots \rangle$. In practice, we can therefore *formally* treat $\psi_\theta^a(\tau)$ in the source term as having the same regularity properties (continuity and differentiability) as any time-dependent quantum-mechanical observable. Moreover, we can use the on-shell Heisenberg evolution, namely

$$\mathrm{Ad}_{H_\theta^a(t)} \sim -\mathrm{i} \frac{\partial}{\partial t}, \tag{35}$$

see the Appendix A, to obtain

$$\psi_\theta^a(\tau) = \mathrm{e}^{-\theta^a(\tau)\frac{\partial}{\partial\tau}} \psi^a(\tau). \tag{36}$$

Introducing the following shorthand notations for the above derivative expansion,

$$
\begin{aligned}
f(t \oplus \epsilon) &\equiv \mathrm{e}^{\epsilon\frac{\partial}{\partial t}} f(t) = f(t) + \epsilon \partial_t f(t) + \tfrac{1}{2!} \epsilon^2 \partial_t^2 f(t) + \ldots \\
f(t \ominus \epsilon) &\equiv \mathrm{e}^{-\epsilon\frac{\partial}{\partial t}} f(t) = f(t) - \epsilon \partial_t f(t) + \tfrac{1}{2!} \epsilon^2 \partial_t^2 f(t) + \ldots
\end{aligned}
\tag{37}
$$

$\psi_\theta^a(\tau)$ is recast as

$$\psi_\theta^a(\tau) = \psi^a(\tau \ominus \theta^a(\tau)). \tag{38}$$

Using once again that sources are set to zero at the end of any physical computation, we integrate by parts the source term of the generating functional in Eq. (26) and safely discard the resulting boundary terms (provided that the dynamics are not probed at $t = \pm t_0$). Finally, the generating functional is now expressed as

$$Z_\theta[J_\theta^+, J_\theta^-] = \mathscr{Z}(-t_0)^{-1} \int \mathscr{D}[\psi^+, \psi^-] \exp\left(\mathrm{i} S_\theta[\psi^+, \psi^-]\right)$$
$$\times \exp\left( \sum_{a=\pm} \int_{\mathscr{C}_\theta^a} \mathrm{d}\tau \, J_\theta^a(\tau \oplus \theta^a(\tau)) \psi^a(\tau) \right), \tag{39}$$

where the action simply reads (omitting the branch superscripts)

$$S_\theta[\psi^+, \psi^-] = \int_{\mathscr{C}_\theta} \mathrm{d}\tau \, \mathscr{L}[\psi(\tau)], \tag{40}$$

and where we introduced the sources $J_\theta^a(\tau)$, defined on the new complex-time contour, and related to the original real-time sources through

$$J_\theta^a(\tau) \equiv \frac{\partial t^a(\tau)}{\partial \tau} J^a(t^a(\tau)). \tag{41}$$

## 3.2 Equilibrium contour

We now choose to work with the following constant and purely imaginary $\theta^a(t)$:

$$\theta^a(t) = a\mathrm{i}\beta/4 \text{ such that } \tau^a(t) = t + a\mathrm{i}\beta/4 \qquad t \in [-t_0; t_0], \quad a = +, -. \tag{42}$$

This corresponds to the complex-time contour $\mathscr{C}_\beta$ represented in Fig. 3. In the rest of this manuscript, we replace the subscript $\theta$ with the subscript $\beta$ to make it clear that we are working with the particular $\theta^a(t)$ given in Eq. (42). The choice of less symmetric contours, such

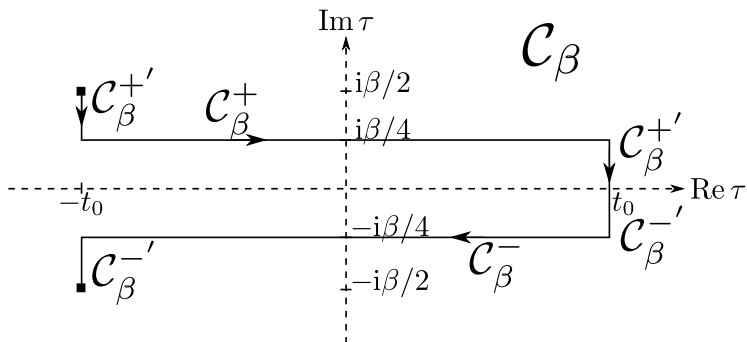

Figure 3: Equilibrium contour, $\mathscr{C}_\beta$, defined in Eq. (42) and used to identify the symmetry of equilibrium dynamics. The black squares mark the boundary condition $\psi^+(-t_0 + \mathrm{i}\beta/2) = \psi^-(-t_0 - \mathrm{i}\beta/2)$.

as the one represented in Fig. 4 (bottom), is also possible but less convenient. Interestingly enough, we note the striking similarity of this "equilibrium" contour $\mathscr{C}_\beta$ with the contour appearing in the thermo-field double formalism [31].

The generating functional reads

$$Z_\beta[J_\beta^+, J_\beta^-] = \mathscr{Z}(-t_0)^{-1} \int \mathscr{D}[\psi^+, \psi^-] \exp\left(\mathrm{i}S_\beta[\psi^+, \psi^-]\right)$$

$$\times \exp\left(\sum_{a=\pm} \int_{\mathscr{C}_\beta^a} \mathrm{d}\tau\, J_\beta^a(\tau \oplus a\mathrm{i}\beta/4)\, \psi^a(\tau)\right), \qquad (43)$$

with the action

$$S_\beta[\psi^+, \psi^-] = \int_{\mathscr{C}_\beta} \mathrm{d}\tau\, \mathscr{L}[\psi(\tau)], \qquad (44)$$

and where the sources on the contour $\mathscr{C}_\beta$ are related to the original real-time sources through

$$J_\beta^a(\tau) = J^a(\tau - a\mathrm{i}\beta/4) \qquad \tau \in \mathscr{C}_\beta^a, \quad a = +, -. \qquad (45)$$

### 3.3 Equilibrium symmetry

Below, we prove that the zero-source generating functional, $Z_\beta[J_\beta^+ = J_\beta^- = 0]$, is invariant under the following transformation of (real) fields:

$$\mathscr{T}_\beta: \ \psi^a(\tau) \mapsto \psi^a(-\tau + a\mathrm{i}\beta/2) \qquad \tau \in \mathscr{C}_\beta^a \cup \mathscr{C}_\beta^{a'}, \quad a = +, -. \qquad (46)$$

For bosonic fields, the transformation includes a field conjugation:

$$\mathscr{T}_\beta: \ \begin{cases} \psi^a(\tau) \mapsto \psi^a(-\tau + a\mathrm{i}\beta/2)^* \\ \psi^a(\tau)^* \mapsto \psi^a(-\tau + a\mathrm{i}\beta/2) \end{cases}, \qquad (47)$$

and for fermionic fields, it also includes a change of sign to accommodate for their anti-commuting nature:

$$\mathscr{T}_\beta: \ \begin{cases} \psi^a(\tau) \mapsto -a\, \psi^a(-\tau + a\mathrm{i}\beta/2)^* \\ \psi^a(\tau)^* \mapsto a\, \psi^a(-\tau + a\mathrm{i}\beta/2) \end{cases}. \qquad (48)$$

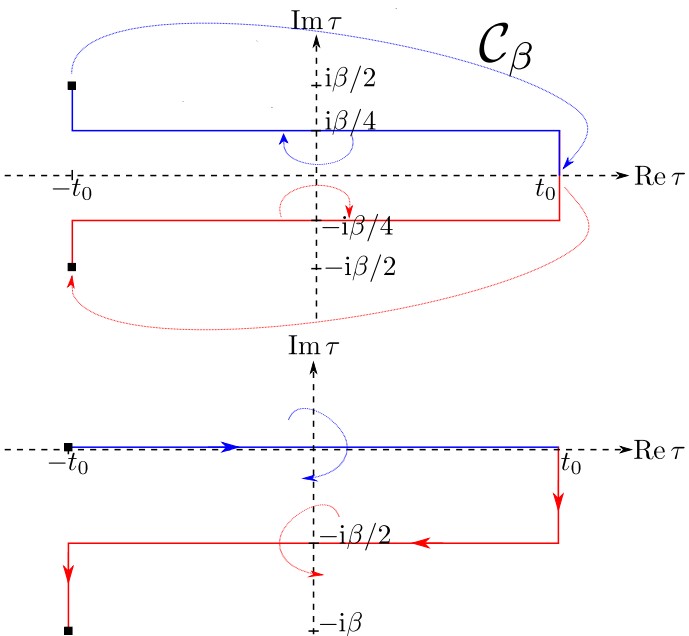

Figure 4: (Top.) Graphical proof of invariance of the equilibrium contour $\mathscr{C}_\beta$ under $\mathscr{T}_\beta$: the blue upper (respectively red lower) contour is invariant under a 180-degree rotation around $\tau = i\beta/4$ (respectively $\tau = -i\beta/4$). The black squares mark the boundary condition $\psi^+(-t_0 + i\beta/2) = \psi^-(-t_0 - i\beta/2)$. The arrows are guides to the eye indicating how specific points transform under $\mathscr{T}_\beta$. (Bottom.) Another possible equilibrium contour, symmetric under a rotation of its real time branch around $\tau = 0$ together with a rotation of its lower branch around $\tau = -i\beta/2$.

Unless otherwise stated, we work with real scalar fields and use the transformation in Eq. (46).

At the level of the "sourced" generating functional, we obtain the relation

$$
Z[J^+(t), J^-(t)] \overset{\text{deform.}}{=} Z_\beta[J_\beta^+(\tau), J_\beta^-(\tau)]
$$
$$
\overset{\mathscr{T}_\beta}{=} Z_\beta[J_\beta^+(-\tau + i\beta/2), J_\beta^-(-\tau - i\beta/2)], \tag{49}
$$

with $J_\beta^a(\tau) \equiv J^a(\tau - ai\beta/4)$ and $\tau \in \mathscr{C}_\beta^a$. The second equality in Eq. (49) is one of the main results of the manuscript. We explore a number of its consequences, namely, fluctuation-dissipation theorems in Sect. 3.4 below.

**Remarks.**

1. The transformation $\mathscr{T}_\beta$ given in Eq. (46) is universal in the sense that it is system independent. In particular, it only depends on a single parameter, the temperature.

2. It is discrete, linear, and involutive, *i.e.* $\mathscr{T}_\beta \circ \mathscr{T}_\beta = \text{Id}$.

3. It is interesting to see $\mathscr{T}_\beta$ as the combination of time reversal with a Kubo-Martin-Schwinger (KMS) transformation [18]

$$
\mathscr{T}_\beta = \mathscr{K}_\beta \circ \text{T}, \tag{50}
$$

where the time-reversal transformation reads, with the notation $\bar{a} \equiv -a$

$$
\text{T}: \begin{cases} \psi^a(\tau) & \mapsto \psi^{\bar{a}}(-\tau) & \text{if } \tau \in \mathscr{C}^a_\beta \\ \psi^a(\tau) & \mapsto \psi^{\bar{a}}(\tau + 2t_0 - ai\beta/2) & \text{if } \operatorname{Re}\tau = -t_0 \quad a = +,-, \\ \psi^a(\tau) & \mapsto \psi^{\bar{a}}(\tau - 2t_0 - ai\beta/2) & \text{if } \operatorname{Re}\tau = t_0 \end{cases} \tag{51}
$$

together with a field conjugation in the case of complex bosonic or fermionic fields. The KMS-like transformation reads

$$
\mathscr{K}_\beta: \begin{cases} \psi^a(\tau) & \mapsto \psi^{\bar{a}}(\tau - a\,i\beta/2) & \text{if } \tau \in \mathscr{C}^a_\beta \\ \psi^a(\tau) & \mapsto \psi^{\bar{a}}(-\tau - 2t_0) & \text{if } \operatorname{Re}\tau = -t_0 \quad a = +,-. \\ \psi^a(\tau) & \mapsto \psi^{\bar{a}}(-\tau + 2t_0) & \text{if } \operatorname{Re}\tau = t_0 \end{cases} \tag{52}
$$

Both T and $\mathscr{K}_\beta$ map the action

$$
iS_\beta[\psi^+, \psi^-] \mapsto \left( iS_\beta[\psi^+, \psi^-] \right)^* , \tag{53}
$$

leaving the zero-source generating functional, $Z_\beta[J^+_\beta = J^-_\beta = 0]$, invariant given that it is a real quantity.

4. In cases in which the system is coupled to an equilibrium environment, the equilibrium symmetry naturally extends to the latter, as already found in Ref. [18] when initial conditions are in the far past and can be neglected. This implies that, when integrating out the environmental degrees of freedom, the corresponding hybridization kernels (or self-energies) that appear in the reduced theory for the non-unitary dynamics of the system are also symmetric under $\mathscr{T}_\beta$.

5. Importantly, the time contour $\mathscr{C}_\beta$ is by itself invariant under $\mathscr{T}_\beta$. Only in this case $\mathscr{T}_\beta$ is a symmetry of the field theory. If one chooses to work with another contour, such as the standard Kadanoff-Baym contour $\mathscr{C}$ represented in Fig. 1, $\mathscr{T}_\beta$ maps a theory where fields are defined on $\mathscr{C}$ to another theory where fields are defined on another contour, hence it is not a field-theoretical symmetry. This is the case for the transformation proposed in Eq. (1) of Ref. [18]. It is only when initial and final times are at infinity such that the initial conditions can be forgotten, that $\mathscr{C}$ and $\mathscr{C}_\beta$ become equivalent (see the discussion in Sect. 3.5), and that Eq. (1) in Ref. [18] becomes a field-theoretical symmetry. This subtlety explains the difference we find with the approach of Ref. [18] when considering the grand-canonical ensemble (*i.e.* in the presence of a chemical potential $\mu$), or when taking the classical limit.

6. $\mathscr{T}_\beta$ is uniquely defined on the contour $\mathscr{C}_\beta$. However, other equivalent contours such as the one represented in Fig. 4 (bottom) are associated with a different equilibrium transformation. In the example of Fig. 4 (bottom), it reads $\psi^+(\tau) \mapsto \psi^+(-\tau)$ for $\tau \in [-t_0, t_0]$ and $\psi^-(\tau) \mapsto \psi^-(-\tau - i\beta)$ for

$$
\tau \in [t_0, t_0 - i\beta/2] \cup [t_0 - i\beta/2, -t_0 - i\beta/2] \cup [-t_0 - i\beta/2, -t_0 - i\beta].
$$

**Proof.**

To prove the relation in Eq. (49), (*i*) we prove the invariance of the action under the transformation $\mathscr{T}_\beta$; (*ii*) we discuss the transformation of the source term; (*iii*) we show the invariance of the integration measure.

(*i*) The action

$$S_\beta[\psi^+, \psi^-] = \left( \int_{-t_0+i\beta/2}^{-t_0+i\beta/4} d\tau + \int_{-t_0+i\beta/4}^{t_0+i\beta/4} d\tau + \int_{t_0+i\beta/4}^{t_0} d\tau \right) \mathscr{L}[\psi^+(\tau)]$$
$$+ \left( \int_{-t_0-i\beta/4}^{-t_0-i\beta/2} d\tau + \int_{t_0-i\beta/4}^{-t_0-i\beta/4} d\tau + \int_{t_0}^{t_0-i\beta/4} d\tau \right) \mathscr{L}[\psi^-(\tau)] \tag{54}$$

transforms as

$$\left( \int_{-t_0+i\beta/2}^{-t_0+i\beta/4} d\tau + \int_{-t_0+i\beta/4}^{t_0+i\beta/4} d\tau + \int_{t_0+i\beta/4}^{t_0} d\tau \right) \mathscr{L}[\psi^+(-\tau+i\beta/2)]$$
$$+ \left( \int_{-t_0-i\beta/4}^{-t_0-i\beta/2} d\tau + \int_{t_0-i\beta/4}^{-t_0-i\beta/4} d\tau + \int_{t_0}^{t_0-i\beta/4} d\tau \right) \mathscr{L}[\psi^-(-\tau-i\beta/2)] \tag{55}$$
$$= - \left( \int_{t_0}^{-t_0+i\beta/4} d\tau + \int_{t_0+i\beta/4}^{-t_0+i\beta/4} d\tau + \int_{-t_0+i\beta/4}^{-t_0+i\beta/2} d\tau \right) \mathscr{L}[\psi^+(\tau)]$$
$$- \left( \int_{t_0-i\beta/4}^{t_0} d\tau + \int_{-t_0-i\beta/4}^{t_0-i\beta/4} d\tau + \int_{-t_0-i\beta/2}^{-t_0-i\beta/4} d\tau \right) \mathscr{L}[\psi^-(\tau)] \tag{56}$$
$$= S_\beta[\psi^+, \psi^-], \tag{57}$$

hence proving that it is invariant. In the second step above, we simply performed a change of (dummy) integration variable, $\tau \mapsto -\tau + ai\beta/2$, and in the last step we exchanged the boundaries of integration.

(*ii*) The source term

$$\sum_{a=\pm} \int_{\mathscr{C}_\beta^a} d\tau \, J_\beta^a(\tau \oplus ai\beta/4) \psi^a(\tau) \tag{58}$$

transforms as

$$\sum_{a=\pm} \int_{\mathscr{C}_\beta^a} d\tau \, J_\beta^a(-\tau + ai\beta/2 \oplus ai\beta/4) \psi^a(\tau). \tag{59}$$

(*iii*) The functional integration measure

$$\int \mathscr{D}[\psi^+, \psi^-] \tag{60}$$

is invariant under the transformation $\mathscr{T}_\beta$ for two reasons: $\mathscr{T}_\beta$ is involutive and it therefore has unit Jacobian, and the integration domain corresponds, before and after the transformation, to integrating over real-valued fields.

## 3.4 Ward identities and fluctuation-dissipation theorem

The symmetry of the action under $\mathscr{T}_\beta$ translates to discrete Ward-Takahashi identities at the level of correlation functions. Let us investigate the immediate consequences of this symmetry on the Keldysh Green's functions $iG^{ab}(t, t') \equiv \langle \psi^a(t) \psi^b(t') \rangle$ with $t, t' \in [-t_0, t_0]$ and $a, b = +, -$. In order to express them in our modified formalism, we use the equivalence

$$\langle \psi^a(t) \ldots \rangle \longleftrightarrow a \frac{\delta Z}{\delta J^a(t)} \bigg|_{J^a=0} \overset{\text{deform.}}{=} a \frac{\delta Z_\beta}{\delta J_\beta^a(\tau)} \bigg|_{J_\beta^a=0} \longleftrightarrow \langle \psi^a(\tau^a \ominus ai\beta/4) \ldots \rangle_\beta, \tag{61}$$

with $J_\beta^a(\tau) = J^a(\tau - ai\beta/4)$ and $\tau^a = t + ai\beta/4$. $\langle\dots\rangle$ indicates the average with respect to $Z[J^+ = J^- = 0]$, the zero-source Kadanoff-Baym generating functional expressed in Eq. (5), and $\langle\dots\rangle_\beta$ is the average with respect to $Z_\beta[J_\beta^+ = J_\beta^- = 0]$ given in Eq. (43). This yields

$$iG^{ab}(t,t') = \langle \psi^a(t + ai\beta/4 \ominus ai\beta/4)\,\psi^b(t' + bi\beta/4 \ominus bi\beta/4)\rangle_\beta \,. \tag{62}$$

Using the analyticity of Green's functions, followed by the symmetry under $\mathcal{T}_\beta$ and, later, the commutativity of real fields, we obtain the relation

$$iG^{ab}(t \oplus ai\beta/4, t' \oplus bi\beta/4) = \langle \psi^a(t + ai\beta/4)\psi^b(t' + bi\beta/4)\rangle_\beta \tag{63}$$
$$\overset{\mathcal{T}_\beta}{=} \langle \psi^a(-t + ai\beta/4)\psi^b(-t' + bi\beta/4)\rangle_\beta = iG^{ba}(-t' \oplus bi\beta/4, -t \oplus ai\beta/4),$$

which reads, equivalently,

$$G^{ab}(t,t') = e^{-i\beta/2\,(a\partial_t + b\partial_{t'})}\, G^{ba}(-t',-t)\,. \tag{64}$$

Using the time-translational invariance of equilibrium dynamics, which makes Green's functions depend on the time difference $t - t'$, we get the following Ward-Takahashi identities corresponding to the discrete equilibrium symmetry:

$$G^{ab}(t) = e^{-i\beta(a-b)/2\,\partial_t}\, G^{ba}(t)\,. \tag{65}$$

In particular, setting $a = -$, $b = +$, this reads

$$G^{-+}(t) = e^{i\beta\partial_t}\, G^{+-}(t)\,. \tag{66}$$

Rather than working in the $\pm$-basis, it is more natural to perform a Keldysh rotation and work with the Keldysh ($G^K$), retarded ($G^R$), and advanced ($G^A$) Green's functions. They are related to the greater and lesser Green's functions through

$$G^R(t,t') - G^A(t,t') = G^{-+}(t,t') - G^{+-}(t,t')\,, \tag{67}$$

$$G^K(t,t') = \frac{1}{2}\big[G^{-+}(t,t') + G^{+-}(t,t')\big]\,. \tag{68}$$

Expressed in the Keldysh basis, Eq. (66) is the quantum fluctuation-dissipation theorem,

$$2\sinh\left(i\frac{\beta}{2}\frac{\partial}{\partial t}\right)G^K(t) = \cosh\left(i\frac{\beta}{2}\frac{\partial}{\partial t}\right)\big[G^R(t) - G^A(t)\big]\,, \tag{69}$$

which relates the linear response (right-hand side), to the equilibrium correlation function (left-hand side). This real-time expression of the quantum FDT, is one of the important immediate consequences of the symmetry unveiled above. Note that while we used the invariance of the action under $\mathcal{T}_\beta = \mathcal{K}_\beta \circ T$, the sole invariance of the zero-source generating functional $Z_\beta[J_\beta^+ = J_\beta^- = 0]$ under $\mathcal{K}_\beta$ is enough to prove the FDT.

**Bosonic and fermionic versions.** In the case of bosonic or fermionic fields, the derivation closely follows the previous steps. For bosons, one obtains the same expression as in Eq. (69), while for fermions, the anti-commutation of the fields yields

$$2\cosh\left(i\frac{\beta}{2}\frac{\partial}{\partial t}\right)G^K(t) = \sinh\left(i\frac{\beta}{2}\frac{\partial}{\partial t}\right)\big[G^R(t) - G^A(t)\big]\,. \tag{70}$$

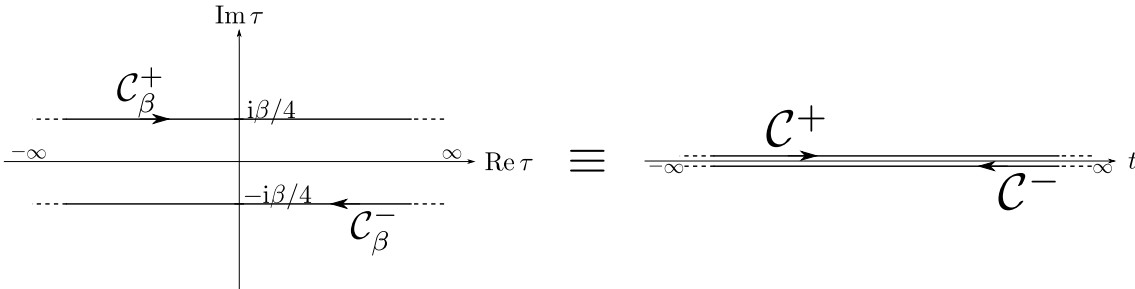

Figure 5: In the limit $t_0 \to \infty$, the generating functional on the contour $\mathscr{C}_\beta$ is equivalent to a conventional Keldysh generating functional on the infinite real-time contour $\mathscr{C}$.

Note that when working with initial conditions in the grand-canonical rather than canonical ensemble, *i.e.* $\rho(-t_0) \sim e^{-\beta(H(-t_0)-\mu N)}$, the corresponding equilibrium generating functional can be obtained rather straightforwardly and is quite similar to the one in Eq. (43). See also the discussion around Eq. (31). We simply mention that its source term reads

$$\sum_{a=\pm} \int_{\mathscr{C}_\beta^a} d\tau \left( J_\beta^a(\tau \oplus ai\beta/4)^* e^{a\beta\mu/4} \psi^a(\tau) + e^{-a\beta\mu/4} \psi^a(\tau)^* J_\beta^a(\tau \oplus ai\beta/4) \right). \tag{71}$$

Importantly, the equilibrium transformation $\mathscr{T}_\beta$ that leaves the zero-source generating functional invariant is unchanged. In particular, $\mathscr{T}_\beta$ does not involve the chemical potential $\mu$. This is different from the transformation $\mathscr{T}_\beta$ proposed in Ref. [18]. Ultimately, the constant factors $e^{\pm\beta\mu/4}$ in the above expression enter the fluctuation theorems in Eqs. (69) and (70) through simply replacing the derivative operator

$$i\frac{\partial}{\partial t} \mapsto i\frac{\partial}{\partial t} - \mu. \tag{72}$$

## 3.5 Schwinger-Keldysh limit

When systems thermalize (either by being in contact with a thermal bath, or self-thermalizing through many-body interactions), their subsequent dynamics cannot depend on the initial conditions. In the limit $-t_0 \mapsto -\infty$, one can therefore safely drop the Matsubara-like imaginary-time branches $\mathscr{C}_\beta^{+'} \cup \mathscr{C}_\beta^{-'}$ from the path integral. This yields a time contour that is simply composed of infinite forward and backward branches. Their vertical position in the complex plane is now irrelevant, making our equilibrium contour completely equivalent to the so-called Schwinger-Keldysh real-time contour, see Fig. 5.

Given the time-independence of the Hamiltonian, and the infinite time axis, it is convenient to work with the Fourier modes of the fields, $\psi(\omega)$, with the correspondence $\psi(\tau) = \int \frac{d\omega}{2\pi} e^{-i\omega\tau} \psi(\omega)$. In this language, the transformation $\mathscr{T}_\beta$ that leaves the action invariant reads

$$\mathscr{T}_\beta : \ \psi^a(\omega) \mapsto e^{-a\beta\omega/2} \psi^a(-\omega) \qquad \omega \in \mathbb{R}, \quad a = +,-. \tag{73}$$

which reads just as in Ref. [18]. In the Keldysh basis defined by the rotation

$$\psi(\omega) \equiv \frac{\psi^+(\omega) + \psi^-(\omega)}{2}, \qquad \hat{\psi}(\omega) \equiv \psi^+(\omega) - \psi^-(\omega), \tag{74}$$

it becomes

$$\mathscr{T}_\beta : \begin{cases} \psi(\omega) & \mapsto \cosh(\beta\omega/2)\psi(-\omega) - 1/2 \sinh(\beta\omega/2)\hat{\psi}(-\omega) \\ \hat{\psi}(\omega) & \mapsto -2\sinh(\beta\omega/2)\psi(-\omega) + \cosh(\beta\omega/2)\hat{\psi}(-\omega) \end{cases} \tag{75}$$

for $\omega \in \mathbb{R}$. The transformation $\mathscr{T}_\beta$ can be seen as a hyperbolic rotation in Keldysh space with an energy dependent angle $\beta\omega/2$ (*i.e.* a KMS transform) combined with a field conjugation (*i.e.* time reversal). In accordance with Eq. (70), the resulting quantum fluctuation-dissipation theorem reads

$$G^K(\omega) = \mathrm{i}\coth(\beta\omega/2)\,\mathrm{Im}\,G^R(\omega). \tag{76}$$

The real part of the retarded Green's function can be computed using the Kramers-Kronig relation.

# 4 Non-equilibrium dynamics: breaking the symmetry

In this Section, we relax the assumption of equilibrium dynamics made in Sect. 3, and consider cases in which the system is driven out of equilibrium by time-dependent potential forces: $H(t) = H(\lambda(t))$ where $\lambda(t) \in \mathbb{R}$ is a time-dependent protocol controlled externally. Although equilibrium considerations lead us to introduce the complex time re-parametrization $\tau^a(t) \equiv t + a\mathrm{i}\beta/4$, and its corresponding equilibrium contour $\mathscr{C}_\beta$ represented in Fig. 3, we continue working in this framework since it will prove to be handy.

There has been a long debate in the literature concerning the appropriate definition of work and entropy production along a single non-equilibrium quantum trajectory. Starting from unquestionable definitions of the average work $\langle\mathscr{W}\rangle$, *i.e.* once averaged over all quantum trajectories, different propositions for single-trajectory definitions were put forward. In the case of isolated systems, the conservation of energy yields $\langle\mathscr{W}\rangle = \mathrm{Tr}[\rho(t_0)H(\lambda(t_0))] - \mathrm{Tr}[\rho(-t_0)H(\lambda(-t_0))]$. This lead to propose a definition of work along a single trajectory as a two-time projective measurement of the total energy of the system: $\mathscr{W} = E(\lambda(t_0))_n - E(\lambda(-t_0))_m$ where the $E(\lambda)_n$'s are the instantaneous eigenvalues of $H(\lambda)$. While successful in many ways, *e.g.* in deriving the quantum work fluctuation relations, this prevents from giving a concrete meaning to work until the second projective measurement is performed. Other proposals started from systems which can exchange heat with an environment. In these cases, the first law yields $\langle\mathscr{W}\rangle = \int_{-t_0}^{t_0} \mathrm{d}t\,\partial_t\lambda(t)\,\mathrm{Tr}[\rho(t)\partial_\lambda H(\lambda(t))]$. In particular, this approach recently inspired Deffner to propose an expression for the quantum entropy production along a single path in phase space [32], see also Ref. [33] for an earlier similar proposition.

Below, we shall follow the same procedure developed in Ref. [9] for classical systems. We identify the symmetry-breaking term that is generated in the transformation of the zero-source generating functional under $\mathscr{T}_\beta$. This quantity turns out to be a fundamental measure of the degree of irreversibility, defined at the level of a single Keldysh trajectory. Considering many limits, we find that it displays all the properties one would expect of a *quantum* version of the dissipated work, if ever such a notion exists. To the best of our knowledge, this is the first time that the framework of stochastic thermodynamics [34], *i.e.* classical thermodynamics at the level of individual trajectories, is successfully generalized to the quantum realm.

## 4.1 Variation of the action under $\mathscr{T}_\beta$

We compute the variation of $Z[J^+ = J^- = 0; \lambda]$, the zero-source non-equilibrium generating functional derived in Eq. (26), under the transformation $\mathscr{T}_\beta$ given in Eq. (46). We recall that

$$Z[J^+ = J^- = 0; \lambda] = \int \mathscr{D}[\psi^+, \psi^-] \exp\left(\beta\mathscr{F}_\lambda(-t_0) + \mathrm{i}S_\beta[\psi^+, \psi^-; \lambda]\right), \tag{77}$$

where the initial partition function has been written in terms of the free energy $\mathscr{F}_\lambda(-t_0) = -\beta^{-1} \ln \mathscr{Z}_\lambda(-t_0)$. The action can be rewritten as the sum to two terms

$$S_\beta[\psi^+, \psi^-; \lambda] = \overline{S}_\beta[\psi^+, \psi^-; \lambda] + \widetilde{S}_\beta[\psi^+, \psi^-; \lambda], \tag{78}$$

with

$$\overline{S}_\beta[\psi^+, \psi^-; \lambda] \equiv \int_{\mathscr{C}_\beta} d\tau \, \mathscr{L}[\psi(\tau); t(\tau)], \tag{79}$$

$$\widetilde{S}_\beta[\psi^+, \psi^-; \lambda] \equiv \int_{-t_0}^{t_0} dt \, \widetilde{\mathscr{L}}_\beta^+[\psi^+(\tau^+(t)); t] - \widetilde{\mathscr{L}}_\beta^-[\psi^-(\tau^-(t)); t], \tag{80}$$

where

$$\widetilde{\mathscr{L}}_\beta^a[\psi^a(\tau^a(t)); t] = -a\, i\beta/4 \int_0^1 dx \, \langle \psi^a(t + ai\beta/4)| \ldots$$
$$\ldots e^{ax\beta/4 \mathrm{Ad}_{H(\lambda(t))}} \, \partial_t H(\lambda(t))|\psi^a(t + ai\beta/4)\rangle. \tag{81}$$

$(i)$ Let us first address the contribution of $\overline{S}_\beta[\psi^+, \psi^-; \lambda]$ to the variation of $Z[J^+ = J^- = 0; \lambda]$. The Lagrangian $\mathscr{L}$ can be separated into time-independent and time-dependent components, $\mathscr{L}_0$ and $\mathscr{L}_1$ respectively,

$$\mathscr{L}[\psi(t); \lambda(t)] = \mathscr{L}_0[\psi(t)] + \mathscr{L}_1[\psi(t); \lambda(t)]. \tag{82}$$

The terms involving $\mathscr{L}_0[\psi(t)]$ are automatically invariant under $\mathscr{T}_\beta$, see Sect. 3. Therefore, we are left with the transformation of $\mathscr{L}_1$,

$$\int_{\mathscr{C}_\beta^a} d\tau \, \mathscr{L}_1[\psi^a(\tau); \lambda(t(\tau))] \overset{\mathscr{T}_\beta}{\mapsto} \int_{\mathscr{C}_\beta^a} d\tau \, \mathscr{L}_1[\psi^a(\tau); \bar{\lambda}(t(\tau))], \tag{83}$$

where we introduced the time-reversed protocol $\bar{\lambda}(t) \equiv \lambda(-t)$. Altogether, we obtain

$$\overline{S}_\beta[\mathscr{T}_\beta \psi^+, \mathscr{T}_\beta \psi^-; \lambda] = \overline{S}_\beta[\psi^+, \psi^-; \bar{\lambda}]. \tag{84}$$

$(ii)$ Let us now address the contribution of $\widetilde{S}_\beta[\psi^+, \psi^-; \lambda]$ to the variation of $Z[J^+ = J^- = 0; \lambda]$. Under $\mathscr{T}_\beta$, it transforms as

$$-i\beta/4 \sum_{a=\pm} \int_{-t_0}^{t_0} dt \int_0^1 dx \, \langle \psi^a(-t + ai\beta/4)|e^{ax\beta/4 \mathrm{Ad}_{H(\lambda(t))}} \, \partial_t H(\lambda(t))|\psi^a(-t + ai\beta/4)\rangle$$
$$= i\beta/4 \sum_{a=\pm} \int_{-t_0}^{t_0} dt \int_0^1 dx \, \langle \psi^a(t + ai\beta/4)|e^{ax\beta/4 \mathrm{Ad}_{H(\bar{\lambda}(t))}} \, \partial_t H(\bar{\lambda}(t))|\psi^a(t + ai\beta/4)\rangle$$
$$= -\widetilde{S}_\beta[\psi^+, \psi^-; \bar{\lambda}]. \tag{85}$$

Therefore, we find that the action transforms as

$$iS_\beta[\psi^+, \psi^-; \lambda] \overset{\mathscr{T}_\beta}{\mapsto} iS_\beta[\psi^+, \psi^-; \bar{\lambda}] - \Sigma_\beta[\psi^+, \psi^-; \bar{\lambda}], \tag{86}$$

where, for reasons that will soon become evident, we introduced the *real* quantity

$$\Sigma_\beta[\psi^+, \psi^-; \lambda] = 2i\widetilde{S}_\beta[\psi^+, \psi^-; \lambda]. \tag{87}$$

Altogether, adding the contribution of the initial partition function, the exponent of $Z[J^+ = J^- = 0; \lambda]$ in Eq. (77) transforms as

$$\beta \mathscr{F}_\lambda(-t_0) + iS_\beta[\psi^+, \psi^-; \lambda] \overset{\mathscr{T}_\beta}{\mapsto} \beta \mathscr{F}_{\bar{\lambda}}(-t_0) + iS_\beta[\psi^+, \psi^-; \bar{\lambda}]$$
$$+ \beta \Delta \mathscr{F}_{\bar{\lambda}} - \Sigma_\beta[\psi^+, \psi^-; \bar{\lambda}], \tag{88}$$

with $\Delta \mathscr{F}_\lambda \equiv \mathscr{F}_\lambda(t_0) - \mathscr{F}_\lambda(-t_0) = -\Delta \mathscr{F}_{\bar{\lambda}}$.

**Remarks.**

1. In the absence of a non-equilibrium drive (*e.g.* for $\lambda(t) = \bar{\lambda}(t) = $ constant), the transformation in Eq. (88) naturally boils down to the symmetry of the action discussed in Sect. 3:

$$\beta\mathscr{F}(-t_0) + iS_\beta[\psi^+, \psi^-] \overset{\mathscr{T}_\beta}{\mapsto} \beta\mathscr{F}(-t_0) + iS_\beta[\psi^+, \psi^-]. \tag{89}$$

2. The first two terms in the right-hand side of Eq. (88) directly supersede their equilibrium counterparts recalled in Eq. (89). They correspond to the expression of $Z[J^+ = J^- = 0; \bar{\lambda}]$, *i.e.* to the dynamics of the same system under a time-reversed protocol $\bar{\lambda}(t) = \lambda(-t)$.

3. The last two terms in the right-hand side of Eq. (88), namely $\beta\Delta\mathscr{F}_{\bar{\lambda}} - \Sigma_\beta[\psi^+, \psi^-; \bar{\lambda}]$, do not have an equilibrium counterpart. They are symmetry-breaking terms generated by the transformation $\mathscr{T}_\beta$, and more precisely by the time reversal T (recall that $\mathscr{T}_\beta = \mathscr{K}_\beta \circ T$). $\beta\Delta\mathscr{F}_{\bar{\lambda}}$ simply corresponds to the free energy difference between the initial and the final "virtual" equilibrium states. We characterize $\Sigma_\beta[\psi^+, \psi^-; \bar{\lambda}]$ below.

4. The classical limit of the transformation in Eq. (88) is computed in Eq. (133).

5. Re-writing the transformation $\mathscr{T}_\beta$ as the combination of time reversal with a KMS transformation, $\mathscr{T}_\beta = \mathscr{K}_\beta \circ T$ [see Eq. (50)], it is interesting to note that $\mathscr{K}_\beta$ has the same effect as in equilibrium:

$$iS_\beta[\psi^+, \psi^-; \lambda] \overset{\mathscr{K}_\beta}{\mapsto} \left(iS_\beta[\psi^+, \psi^-; \lambda]\right)^*. \tag{90}$$

### 4.2 Symmetry-breaking term

While the physical interpretation of $\beta\Delta\mathscr{F}_\lambda$ in Eq. (88) is trivial, the physical content of the second symmetry-breaking term, $\Sigma_\beta[\psi^+, \psi^-; \lambda]$, has yet to be discussed. It is a real functional of the trajectories of the fields $\psi^+$ and $\psi^-$, which reads

$$
\begin{aligned}
\Sigma_\beta[\psi^+, \psi^-; \lambda] = \beta/2 \int_{-t_0}^{t_0} dt \int_0^1 dx \Big\{ & \langle\psi^+(t+i\beta/4)|e^{x\beta/4\,\mathrm{Ad}_{H(\lambda(t))}}\,\partial_t H(\lambda(t))|\psi^+(t+i\beta/4)\rangle \\
& + \langle\psi^-(t-i\beta/4)|e^{-x\beta/4\,\mathrm{Ad}_{H(\lambda(t))}}\,\partial_t H(\lambda(t))|\psi^-(t-i\beta/4)\rangle \Big\} \\
= \beta/2 \sum_a \int_{-t_0}^{t_0} dt \int_0^1 dx\, & \langle\psi^a(t+i\beta/4)| \cosh\left(\frac{x\beta}{4}\,\mathrm{Ad}_{H(\lambda(t))}\right) \\
& \times \partial_t H(\lambda(t))|\psi^a(t+i\beta/4)\rangle.
\end{aligned}
\tag{91}
$$

In the second line, we made use of the assumption that the Hamiltonian is even under the time-reversal operator, *i.e.* $\Theta H(t)\Theta = H(t)$, hence $M = e^{-x\beta/4\,\mathrm{Ad}_{H(\lambda(t))}}\,\partial_t H(\lambda(t))$ is also even, ensuring that $\langle\psi|M|\psi\rangle$ is a real quantity where $M$ can therefore be replaced by its Hermitian component: $\langle\psi|M|\psi\rangle = \langle\psi|(M + M^\dagger)/2|\psi\rangle$. In the conventional Kadanoff-Baym formulation, *i.e.* deforming back the formalism to the original contour $\mathscr{C}$ and therefore dropping the subscript $\beta$, $\Sigma_\beta[\psi^+, \psi^-; \lambda]$ corresponds to

$$\Sigma[\psi^+, \psi^-; \lambda] = \sum_{a=\pm} a \int_{-t_0}^{t_0} dt\, \langle\psi^a(t)|\dot{\sigma}^a(t)|\psi^a(t)\rangle, \tag{92}$$

where we introduced the Hermitian rate operator

$$\dot{\sigma}^a(t) \equiv a\beta/2 \, \partial_t \lambda(t) \int_0^1 dx \, \cosh\left(\frac{x\beta}{4} \mathrm{Ad}_{H(\lambda(t))}\right) \partial_\lambda H(\lambda(t)) \tag{93}$$

$$= -a \, e^{\beta/4 H(t)} \partial_t \left[e^{-\beta/2 H(t)}\right] e^{\beta/4 H(t)}, \tag{94}$$

where the cosh in Eq. (93) has to be understood as a sum of exponentials and we recall that $e^{\mathrm{Ad}_Y} X = e^Y X e^{-Y}$. We shall see below that, (i) the classical limit of $\Sigma[\psi^+, \psi^-; \lambda]$ corresponds to the work performed by the external protocol $\lambda(t)$ on a given trajectory of the system (cf. Sect. 5.4); (ii) its average over all quantum trajectories, $\langle\Sigma\rangle$, yields the average work only when the process in reversible; (iii) the quantity $\langle\Sigma\rangle - \beta\Delta\mathcal{F}_\lambda$ is a measure of the irreversibility generated in the process (cf. Sect. 4.3); (iv) the statistics of $\Sigma[\psi^+, \psi^-; \lambda]$ obey the fluctuation relations.

This leads us to conclude that the functional $\Sigma[\psi^+, \psi^-; \lambda] - \beta\Delta\mathcal{F}_\lambda$ is a measure of irreversibility per individual Keldysh trajectory, and can be seen as a quantum generalization of the notion of dissipated work. Although we prefer to remain cautious before giving a name to the operator $\dot{\sigma}^a(t)$ introduced in Eq. (94), it surely plays a role physically equivalent to the long-sought entropy production rate operator.

**Average value.** Performing the average over all trajectories, after a rotation back to the original basis, and the use of the analytic properties of time-dependent observables, we obtain

$$\langle\Sigma\rangle = \beta \int_{-t_0}^{t_0} dt \, \partial_t \lambda(t) \int_0^1 dx \left[ \cos(x\beta/4 \, \partial_t) \partial_\lambda \langle\mathcal{H}(t; \lambda)\rangle \right]_{\lambda=\lambda(t)}, \tag{95}$$

where $\langle\mathcal{H}(t; \lambda(t))\rangle = \mathrm{Tr}[\rho(t) H(\lambda(t))]$ is the average energy at time $t$. This expression for the average measure of irreversibility is one of the main results of this manuscript. It is a true physical quantity which does not involve prior knowledge of our formalism, and which can, in principle, be measured in experiments.

For example, if the time-dependent Hamiltonian is of the form $H(t) = H_0 + \lambda(t) V$ and $\lambda(\pm t_0) = 0$, this reads

$$\langle\Sigma\rangle = -4 \int_{-t_0}^{t_0} dt \, \lambda(t) \sin(\beta/4 \, \partial_t) \langle V(t)\rangle \tag{96}$$

$$= \beta \langle\mathcal{W}\rangle - \frac{\beta^3}{96} \int_{-t_0}^{t_0} dt \, \partial_t \lambda(t) \partial_t^2 \langle V(t)\rangle + \mathcal{O}\left(\beta^5 \partial_t^5 \lambda(t)\right). \tag{97}$$

In the second line, we displayed the first correction to the average work defined as $\langle\mathcal{W}\rangle = \int_{-t_0}^{t_0} dt \, \partial_t \lambda(t) \mathrm{Tr}[\rho(t) \partial_\lambda H(\lambda(t))]$. Aside boundary terms, the higher corrections involve an infinite series of the odd derivatives of $\lambda(t)$.

**Reversible process.** In case of a slowly-varying external protocol, if the system is able to relax fast enough such that its average energy depends on time only through the control parameter, i.e. $\langle\mathcal{H}(\lambda(t))\rangle$, then $\langle\Sigma\rangle$ in Eq. (95) simply reduces to the average quantum-mechanical work (in units of $\beta$):

$$\langle\Sigma\rangle = \beta \int_{-t_0}^{t_0} dt \, \partial_t \lambda(t) \partial_\lambda \langle\mathcal{H}(\lambda(t))\rangle = \beta \langle\mathcal{W}\rangle. \tag{98}$$

Note that for an infinitely large many-body system, the above assumptions typically require the system to be coupled to a fast-relaxing heat bath. Consider, for example, the interacting

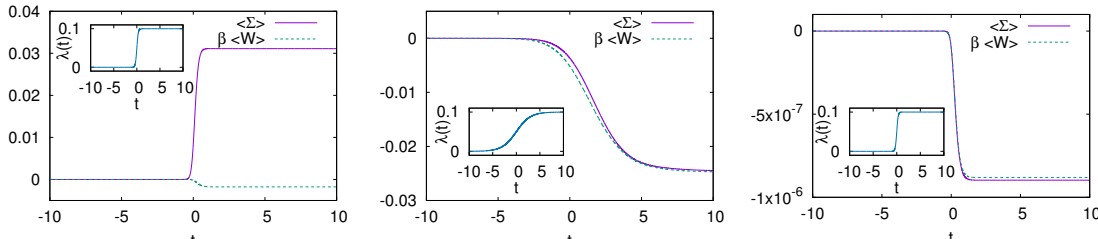

Figure 6: Time evolution of $\langle\Sigma(t)\rangle$ and $\beta\langle\mathscr{W}(t)\rangle$ for the driven two-level system described around Eq. (99). (a) The parameters are $\beta^{-1} = 0.1$, $t_\lambda = 0.3$, and $\lambda_0 = 0.1$. (b) Slow process: same parameters as in (a) but $t_\lambda = 3$. (c) High temperature: same parameters as in (a) but $\beta^{-1} = 10$. The insets represent the time-dependent drive protocol $\lambda(t)$.

electrons of a Hubbard model accelerated by a constant electric field [35, 36]. Clearly, in the absence of a heat bath, even the tiniest electric field will make the average energy monotonically increase in time (until an infinite effective temperature steady state is eventually reached). Reversing or turning off the electric field will not reduce the average energy that has accumulated in the system.

**Simple example: driven two-level system.**   To illustrate the physical content of the novel quantity $\Sigma$, we consider a simple quantum-mechanical system driven out of equilibrium and we follow its dynamics by means of exact numerical integration. We choose a two-level system, prepared in thermal equilibrium at temperature $\beta^{-1}$, and evolved with respect to the Hamiltonian

$$H(t) = \frac{1}{2}\sigma^z + \frac{1}{2}\lambda(t)\sigma^x, \tag{99}$$

where $\sigma^x$ and $\sigma^z$ are the usual Pauli matrices. The time-dependent protocol $\lambda(t) = \lambda_0[1 + \exp(-2t/t_\lambda)]^{-1}$, represented in the inset of Fig. 6, continuously passes from 0 to $\lambda_0 > 0$ on a timescale set by $t_\lambda$. In Fig. 6, we represent the evolution of $\langle\Sigma(t)\rangle = \int_{-t_0}^t dt' \, \text{Tr}\big[\dot\sigma(t')\rho(t')\big]$ where the density matrix $\rho(t)$ is obtained by solving the von Neumann equation of motion, $\partial_t\rho(t) = -i[H(t), \rho(t)]$, with the Gibbs-Boltzmann initial condition $\rho(-t_0) = e^{-\beta H(-t_0)}/\mathscr{Z}$. For comparison, we also plot the average work $\beta\langle\mathscr{W}(t)\rangle = \beta\int_{-t_0}^t dt' \, \text{Tr}[\partial_t H(t)\rho(t)]$, see also Eq. (97). We also check that for a very-slow protocol, or at high temperatures, $\langle\Sigma(t)\rangle$ reduces to the average work $\beta\langle\mathscr{W}(t)\rangle$.

## 4.3   Quantum fluctuation theorems

Beyond its average value, more information on $\Sigma_\beta[\psi^+, \psi^-; \lambda]$ defined in Eq. (91) – or equivalently on $\Sigma[\psi^+, \psi^-; \lambda]$ in Eq. (92) – can be obtained by studying its asymmetric statistics. These are the so-called fluctuation theorems. Within our approach, they simply and naturally result from the generation of symmetry-breaking terms when the zero-source generating functional is transformed under $\mathscr{T}_\beta$. While quantum fluctuation theorems were already associated to the breaking of time-reversal symmetry [37], we believe it is the first time that a proof is given using a path-integral formalism. The main difference with the previous quantum fluctuation theorems is that we do not rely on a two-time measurement projecting the system onto eigenstates at the initial and final time.

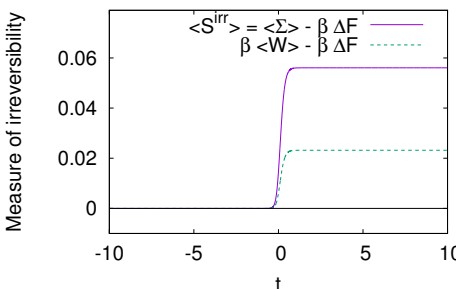

Figure 7: Time evolution of the quantum measure of irreversibility, $\langle \mathscr{S}^{\mathrm{irr}}(t) \rangle = \langle \Sigma(t) \rangle - \beta \Delta \mathscr{F}$, see Eq. (102), for the driven two-level system defined around Eq. (99). The parameters are the same as in Fig. 6(a) ($T = 0.1, t_\lambda = 0.3, \lambda_0 = 0.1$). The classical definition of entropy creation, $\beta \langle \mathscr{W}(t) \rangle - \beta \Delta F$, is plotted for comparison.

**Jarzynski equality.** Let us consider the trivial identity $\langle 1 \rangle_\beta = 1$ and transform its left-hand side under $\mathscr{T}_\beta$. Using the relation in Eq. (88), we immediately obtain the quantum Jarzynski equality [20, 21, 38],

$$\mathrm{e}^{\beta \Delta \mathscr{F}_\lambda} \langle \mathrm{e}^{-\Sigma_\beta [\psi^+, \psi^-; \lambda]} \rangle_\beta = 1, \tag{100}$$

which also reads, back in the conventional Kadanoff-Baym formulation,

$$\mathrm{e}^{\beta \Delta \mathscr{F}_\lambda} \langle \mathrm{e}^{-\Sigma [\psi^+, \psi^-; \lambda]} \rangle = 1. \tag{101}$$

Conjecturing that the quantity $P_\lambda(\Sigma) \equiv \langle \delta(\Sigma[\psi^+, \psi^-; \lambda] - \Sigma) \rangle$ is an appropriate (*i.e.* normalized non-negative) probability distribution, the Jensen's inequality[4] applied on Eq. (101) yields $\langle \Sigma \rangle - \beta \Delta \mathscr{F}_\lambda \geq 0$. The equality is achieved for reversible processes: $\beta \langle \mathscr{W} \rangle^{\mathrm{rev}} - \beta \Delta \mathscr{F}_\lambda = 0$. Therefore, in general, the positive quantity

$$\langle \mathscr{S}^{\mathrm{irr}} \rangle \equiv \langle \Sigma \rangle - \beta \Delta \mathscr{F}_\lambda \geq 0 \tag{102}$$

is a measure of the average irreversibility generated by the time-dependent protocol.

To illustrate this novel measure of irreversibility, let us consider the two-level system introduced around Eq. (99). The time evolution of $\langle \mathscr{S}^{\mathrm{irr}}(t) \rangle$ is numerically computed and displayed in Fig. 7. As expected, it is a positive quantity which reduces to the classical creation of entropy, $\beta \langle \mathscr{W}(t) \rangle - \beta \Delta \mathscr{F}$, in the limit of a reversible process or in the high-temperature regime.

**Crooks fluctuation theorem.** Let us consider the probability

$$P_\lambda(\Sigma) \equiv \langle \delta(\Sigma_\beta [\psi^+, \psi^-; \lambda] - \Sigma) \rangle_\beta, \tag{103}$$

that $\Sigma_\beta [\psi^+, \psi^-; \lambda]$ be $\Sigma$ under the protocol $\lambda$. When transformed under $\mathscr{T}_\beta$, it immediately yields a Crooks fluctuation theorem,

$$P_\lambda(\Sigma) = \mathrm{e}^{\Sigma - \Delta \mathscr{F}_\lambda} P_{\bar{\lambda}}(-\Sigma), \tag{104}$$

---

[4]Jensen's inequality states that for a convex function $\varphi$, a real-valued function $g$, and a random variable $x$, the expectation values satisfy $\mathbb{E}[\varphi(g(x))] \geq \varphi(\mathbb{E}[g(x)])$. This result can be extended to a Feynman-Jensen inequality in our path-integral formalism as long as the quantity $P_\lambda(\Sigma)$ is an appropriate probability distribution, *i.e.* normalized and non-negative, such that one can take $\mathbb{E}[\dots] = \int \mathrm{d}\Sigma \, P_\lambda(\Sigma) \dots$. Clearly $\int \mathrm{d}\Sigma \, P_\lambda(\Sigma) = 1$. The proof of its non-negativity is easy in the classical case and for reversible processes, limits in which $\dot{\sigma}^a(t) = a\beta/2 \, \partial_t H(t)$ and $P_\lambda(\Sigma)$ boils down to the distribution of work that is discussed in Ref. [39]. We conjecture that $P_\lambda(\Sigma)$ is non-negative in the general case.

where $\bar{\lambda}(t) \equiv \lambda(-t)$ is the time-reversed protocol. Notice that Eq. (104) is different from the well-known quantum work fluctuation theorem that characterizes the statistics of work $\mathcal{W}$, computed (in a closed system) as the energy difference between the final and initial time [19, 22]. In fact, Eq. (104) focuses on a different quantity, $\Sigma$, which does not rely on such a two-time projective measurement.

Let us now introduce the following quantity defined on a single Keldysh trajectory on the contour $\mathscr{C}_\beta$,

$$\mathscr{S}_\beta^{\text{irr}}[\psi^+, \psi^-; \lambda] \equiv \Sigma_\beta[\psi^+, \psi^-; \lambda] - \beta \Delta \mathscr{F}_\lambda \tag{105}$$

that corresponds, on the conventional Kadanoff-Baym contour $\mathscr{C}$, to

$$\mathscr{S}^{\text{irr}}[\psi^+, \psi^-; \lambda] \equiv \Sigma[\psi^+, \psi^-; \lambda] - \beta \Delta \mathscr{F}_\lambda, \tag{106}$$

and is such that its average matches $\langle \mathscr{S}_\beta^{\text{irr}}[\psi^+, \psi^-; \lambda] \rangle_\beta = \langle \Sigma \rangle - \beta \Delta \mathscr{F}_\lambda \equiv \langle \mathscr{S}^{\text{irr}} \rangle$. It is straightforward to derive a fluctuation relation for this fluctuating quantity:

$$P_\lambda(\mathscr{S}^{\text{irr}}) = e^{\mathscr{S}^{\text{irr}}} P_{\bar{\lambda}}(-\mathscr{S}^{\text{irr}}). \tag{107}$$

This relates the amount of irreversibily generated/destroyed between the initial and final time under the protocol $\lambda(t)$ with the one generated/destroyed under the time-reversed protocol. In the absence of a time-dependent protocol, we simply obtain $P(\mathscr{S}^{\text{irr}}) = P(-\mathscr{S}^{\text{irr}}) e^{\mathscr{S}^{\text{irr}}}$, expressing the exponentially small probability that irreversibility decreases on a given trajectory.

**Generalized fluctuation theorems.** The above fluctuation theorems can be systematically generalized to any physical quantity $\langle X \rangle$. The strategy is the following: (*i*) first, start from the conventional path-integral expression of $\langle X \rangle$ in the Kadanoff-Baym formulation; (*ii*) next, deform the Kadanoff-Baym contour $\mathscr{C}$ to the equilibrium contour $\mathscr{C}_\beta$; (*iii*) then, apply the transformation $\mathscr{T}_\beta$; (*iv*) finally, deform back the contour to the original contour $\mathscr{C}$.

For example, let us start with the Green's function $iG_\lambda^{ab}(t, t') \equiv {}_\lambda\langle \psi^a(t)\psi^b(t') \rangle$ where the left subscript $\lambda$ indicates that the average is computed with respect to the protocol $\lambda(t)$. We have

$$_\lambda\langle \psi^a(t)\psi^b(t') \rangle \stackrel{\text{deform.}}{=} {}_\lambda\langle \psi_\beta^a(t)\psi_\beta^b(t') \rangle_\beta \tag{108}$$

$$\stackrel{\mathscr{T}_\beta}{=} e^{\beta \Delta \mathscr{F}_{\bar{\lambda}}} {}_{\bar{\lambda}}\langle \psi_\beta^a(-t)\psi_\beta^b(-t') e^{-\Sigma_\beta[\psi^+, \psi^-; \bar{\lambda}]} \rangle_\beta \tag{109}$$

$$\stackrel{\text{deform.}}{=} e^{\beta \Delta \mathscr{F}_{\bar{\lambda}}} {}_{\bar{\lambda}}\langle \psi^a(-t)\psi^b(-t') e^{-\Sigma[\psi^+, \psi^-; \bar{\lambda}]} \rangle, \tag{110}$$

where in the last two lines, the average is computed with respect to the time-reversed protocol $\bar{\lambda}(t) = \lambda(-t)$. The properties that we used are, in the first line,

$$\psi^a(t) \stackrel{\text{deform.}}{\mapsto} \psi_\beta^a(t) = \langle \psi^a(t + ai\beta/4)| e^{a\beta/4 \text{Ad}_{H(\lambda(t))}} \psi^a | \psi(t + ai\beta/4) \rangle, \tag{111}$$

in the second line,

$$\psi_\beta^a(t) \stackrel{\mathscr{T}_\beta}{\mapsto} \langle \psi^a(-t + ai\beta/4)| e^{a\beta/4 \text{Ad}_{H(\bar{\lambda}(-t))}} \psi^a | \psi(-t + ai\beta/4) \rangle, \tag{112}$$

and in the last line, $\Sigma_\beta[\psi^+, \psi^-, \lambda] \stackrel{\text{deform.}}{\mapsto} \Sigma[\psi^+, \psi^-; \lambda]$, the expression of which is given in Eq. (92).

# 5 Classical limit

We have in mind the case of a classical particle of mass $m$ and coordinate $\psi$ in a time-dependent potential $V(\psi; \lambda(t))$, with Lagrangian $\mathscr{L}[\psi(t); t] = \frac{1}{2}m\partial_t^2\psi(t) - V(\psi(t); \lambda(t))$. The generalization to more complex systems is straightforward. The classical limit of our real scalar field theory is obtained by taking the limit $\hbar \to 0$. Let us therefore re-introduce the factors of $\hbar$ and work in the Keldysh basis, *i.e.* rotate the forward and backward fields to the so-called classical and quantum fields,

$$\psi(t) \equiv \frac{\psi^+(t) + \psi^-(t)}{2}, \qquad \hat{\psi}(t) \equiv \frac{\psi^+(t) - \psi^-(t)}{\hbar}. \tag{113}$$

These fields are coupled to the sources $\hat{J}(t) \equiv [J^+(t) - J^-(t)]/\hbar$ and $J(t) \equiv [J^+(t) + J^-(t)]/2$, respectively. To make the connection with the literature on the Martin-Siggia-Rose-Janssen-DeDominicis (MSRJD) formalism easier, we rename the sources $J(t)$ and $\hat{J}(t)$ with $if(t)$ and $\hat{f}(t)$, respectively.

## 5.1 Generating functional

In the classical limit, the terms of the unithermal Lagrangian that contribute to the action are at most of order $\hbar$, *i.e.*

$$\begin{aligned}
\mathscr{L}_\beta^a[\psi^a(t); t] &= \mathscr{L}[\psi^a(t); t] + ai\beta\hbar/4 \frac{\partial \mathscr{L}[\psi; t]}{\partial t}\bigg|_{\psi = \psi^a(t)} \\
&= \mathscr{L}[\psi(t); t] + a\hbar/2 \int dt'\, i\hat{\psi}(t') \frac{\delta \mathscr{L}[\psi(t); t]}{\delta\psi(t')} + ai\beta\hbar/4 \frac{\partial \mathscr{L}[\psi; t]}{\partial t}\bigg|_{\psi = \psi(t)}.
\end{aligned} \tag{114}$$

The classical limit of the action in Eq. (78), $S_{\text{cl}} \equiv \lim_{\hbar \to 0} \frac{i}{\hbar} S_\beta$, reads

$$\begin{aligned}
S_{\text{cl}}[\psi, \hat{\psi}] = {}&- \beta \frac{H[\psi(-t_0); -t_0] + H[\psi(t_0); t_0]}{2} - \beta/2 \int_{-t_0}^{t_0} dt\, \frac{\partial \mathscr{L}[\psi; t]}{\partial t}\bigg|_{\psi = \psi(t)} \\
&+ \int_{-t_0}^{t_0} dt\, i\hat{\psi}(t) \left( \frac{\partial \mathscr{L}[\psi(t); t]}{\partial \psi(t)} - \frac{d}{dt} \frac{\partial \mathscr{L}[\psi(t); t]}{\partial \partial_t \psi(t)} \right),
\end{aligned} \tag{115}$$

where $H[\psi(t); t]$ is the energy of a given configuration $\psi(t)$ at time $t$. We used the boundary conditions $\hat{\psi}(-t_0) = \hat{\psi}(t_0) = 0$ that follow from those discussed in Sect. 2 in the quantum case. In the last term of the action above, one recognizes the Euler-Lagrange equation of motion. It is strictly enforced by the overall integration over $\hat{\psi}(t)$ which acts as a Lagrange multiplier. The classical generating functional reads

$$\begin{aligned}
Z[f, \hat{f}] = {}&\mathscr{Z}(-t_0)^{-1} \int \mathscr{D}[\psi, \hat{\psi}] \exp\left( S_{\text{cl}}[\psi, \hat{\psi}] \right) \\
&\times \exp \int_{-t_0}^{t_0} dt\, \left( f(t) i\hat{\psi}_\beta(t) + \hat{f}(t)\psi(t) \right),
\end{aligned} \tag{116}$$

where

$$i\hat{\psi}_\beta(t) = i\hat{\psi}(t) + \beta/2\, \partial_t \psi(t). \tag{117}$$

The following classical Kubo formula expresses the linear response function $R(t', t') \equiv \delta\langle\psi(t)\rangle/\delta f(t')\big|_{f=0}$ as a two-point correlation function:

$$R(t, t') = \langle\psi(t)i\hat{\psi}_\beta(t')\rangle_{\text{cl}} = \langle\psi(t)i\hat{\psi}(t')\rangle + \beta/2\, \partial_{t'}\langle\psi(t)\psi(t')\rangle. \tag{118}$$

**Thermal bath.** If the system is in contact with a thermal bath at temperature $\beta^{-1}$ during the whole evolution, the action $S_{\text{cl}}$ is supplemented by a dissipative contribution of the form (here for an additive white noise)

$$S_{\text{diss}} = \eta\beta^{-1} \int_{-t_0}^{t_0} dt \, \big(i\hat{\psi} + \beta/2 \, \partial_t \psi\big)\big(i\hat{\psi} - \beta/2 \, \partial_t \psi\big), \tag{119}$$

with the friction coefficient $\eta \geq 0$.

**Example of a massive particle.** In the simple case of a particle of mass $m$ of coordinate $\psi$ in a time-dependent potential $V(\psi; \lambda(t))$, the energy is $\mathcal{H}[\psi(t); t] = \frac{1}{2}m[\partial_t \psi(t)]^2 + V(\psi(t); \lambda(t))$, and the action reads

$$
\begin{aligned}
S_{\text{cl}}[\psi, \hat{\psi}] = & -\beta \, \frac{H[\psi(-t_0); -t_0] + H[\psi(t_0); t_0]}{2} \\
& - \int_{-t_0}^{t_0} dt \, i\hat{\psi}(t)\big(m\partial_t^2 \psi(t) + V'(\psi(t); \lambda(t))\big) \\
& + \beta/2 \int_{-t_0}^{t_0} dt \, \partial_t \lambda(t) \, \partial_\lambda V(\psi(t); \lambda(t)).
\end{aligned}
\tag{120}
$$

Note that the last term corresponds to half of the total work (in units of $\beta$) performed with the protocol $\lambda(t)$, $\beta\mathcal{W}/2$.

**MSRJD formalism.** The connection with the Martin-Siggia-Rose-Janssen-deDominicis (MSRJD) formalism is realized by performing the following change of variable:

$$i\hat{\psi}(t) \mapsto i\hat{\psi}_{\text{MSR}}(t) - \beta/2 \, \partial_t \psi(t), \tag{121}$$

and by using the analyticity of $\exp(S_{\text{cl}}[\psi, \hat{\psi}])$ to bring back the domain of functional integration of $\hat{\psi}_{\text{MSR}}$ over real fields. We thus obtain the conventional MSRJD generating functional

$$
\begin{aligned}
Z[f, \hat{f}] = \mathscr{Z}(-t_0)^{-1} \int \mathscr{D}[\psi, \hat{\psi}_{\text{MSR}}] \exp\big(S_{\text{MSR}}[\psi, \hat{\psi}_{\text{MSR}}]\big) \\
\times \exp \int_{-t_0}^{t_0} dt \, \big(f(t)i\hat{\psi}_{\text{MSR}}(t) + \hat{f}(t)\psi(t)\big),
\end{aligned}
\tag{122}
$$

with the action

$$
\begin{aligned}
S_{\text{MSR}}[\psi, \hat{\psi}] = & -\beta H[\psi(-t_0); -t_0] \\
& - \int_{-t_0}^{t_0} dt \, i\hat{\psi}_{\text{MSR}}(t)\left(\frac{\partial \mathscr{L}[\psi(t); t]}{\partial \psi(t)} - \frac{d}{dt}\frac{\partial \mathscr{L}[\psi(t); t]}{\partial \, \partial_t \psi(t)}\right).
\end{aligned}
\tag{123}
$$

$\hat{\psi}_{\text{MSR}}$ is usually referred as the "response field" due to the corresponding Kubo formula

$$R(t', t') \equiv \left.\frac{\delta\langle\psi(t)\rangle}{\delta f(t')}\right|_{f=0} = \langle\psi(t)i\hat{\psi}_{\text{MSR}}(t')\rangle_{\text{MSR}}. \tag{124}$$

The equation of motion associated to Eq. (123) is simply Newton's equation. When supplemented with a thermal bath, this rather yields a Langevin-type of equation with extra stochastic and viscous forces acting on $\psi$. For the massive particle in a time-dependent potential, the MSRJD action reads

$$
\begin{aligned}
S_{\text{MSR}}[\psi, \hat{\psi}_{\text{MSR}}] = & -\beta H[\psi(-t_0); -t_0] \\
& - \int_{-t_0}^{t_0} dt \, i\hat{\psi}_{\text{MSR}}(t)\big(m\partial_t^2 \psi(t) + V'(\psi(t); \lambda(t))\big).
\end{aligned}
\tag{125}
$$

## 5.2 Equilibrium symmetry

The classical version of the equilibrium transformation in Eq. (46) reads

$$\mathscr{T}_\beta : \begin{cases} \psi(t) & \mapsto & \psi(-t) \\ \hat{\psi}(t) & \mapsto & \hat{\psi}(-t) \end{cases} \qquad t \in [-t_0, t_0]. \tag{126}$$

Note that, similarly to the quantum case, both $\psi$ and $\hat{\psi}$ remain real after the transformation. Once in the MSRJD language, we recover the equilibrium symmetry discussed in Ref. [9]

$$\mathscr{T}_{\text{eq}} : \begin{cases} \psi(t) & \mapsto & \psi(-t) \\ i\hat{\psi}_{\text{MSR}}(t) & \mapsto & i\hat{\psi}_{\text{MSR}}(-t) + \beta \partial_t \psi(-t) \end{cases} \qquad t \in [-t_0, t_0]. \tag{127}$$

Similarly to the quantum case, the classical version of $\mathscr{T}_\beta$ in Eq. (126) can be seen as the composition of a time-reversal transformation

$$\text{T} : \begin{cases} \psi(t) & \mapsto & \psi(-t) \\ \hat{\psi}(-t) & \mapsto & -\hat{\psi}(-t) \end{cases} \quad \text{with} \quad \mathscr{K}_\beta : \begin{cases} \psi(t) & \mapsto & \psi(t) \\ \hat{\psi}(t) & \mapsto & -\hat{\psi}(t) \end{cases} . \tag{128}$$

Note that in Ref. [18] the transformation (127) is obtained directly as the classical limit of the transformation (Eq. (27) in Ref. [18]), without any additional field transformation. As noticed previously, this is due to the difference in the contour chosen to set up the field theory, see the discussion in Sec. 3.3, and shows the difference between the two approaches.

## 5.3 Fluctuation-dissipation theorem

Similarly to the quantum case, the fluctuation-dissipation theorem [40, 41] is one of the immediate consequences of the equilibrium symmetry of the generating functional. It answers the question: how does the linear response $R(t, t')$ transform under $\mathscr{T}_\beta$? We start from the transformation of $i\hat{\psi}_\beta$ defined in Eq. (117),

$$\begin{aligned} i\hat{\psi}_\beta(t) & \overset{\mathscr{T}_\beta}{\mapsto} & i\hat{\psi}(-t) + \beta/2\, \partial_t \psi(-t) \\ & = & i\hat{\psi}_\beta(-t) + \beta \partial_t \psi(-t). \end{aligned} \tag{129}$$

Therefore $R(t, t')$ transforms as

$$\begin{aligned} R(t, t') = \langle \psi(t) i\hat{\psi}_\beta(t') \rangle & \overset{\mathscr{T}_\beta}{=} & \langle \psi(-t) i\hat{\psi}_\beta(-t) \rangle + \beta \partial_{t'} \langle \psi(-t)\psi(-t') \rangle \\ & = & R(-t, -t') + \beta \partial_{t'} C(-t, -t'). \end{aligned} \tag{130}$$

The proof of the fluctuation-dissipation theorem is immediate after we assume time-translational invariance and we use the causality of the response function. In this way we obtain

$$R(t) = -\vartheta(t)\beta\, \partial_t C(t), \tag{131}$$

where $\vartheta(t)$ is the Heaviside step function. Generalized fluctuation-dissipation theorems, involving responses of more complex quantities to generic linear perturbations, or higher-order multi-time correlation functions, can be derived in a similar fashion [9].

### 5.4 Broken symmetry: fluctuation theorems

In the same spirit as in the quantum case, fluctuation theorems are immediately obtained from the symmetry-breaking terms that are spontaneously generated by the transformation of the action under $\mathcal{T}_\beta$. In the classical limit, the functional $\Sigma_\beta[\psi^+, \psi^-; \lambda]$ identified in Eq. (91) simplifies to the work $\mathcal{W}[\psi; \lambda]$ performed by the protocol $\lambda(t)$ along a trajectory $\psi(t)$,

$$\lim_{\hbar \to 0} \Sigma_\beta[\psi^+, \psi^-; \lambda] = \beta \int \mathrm{d}t \, \partial_t \lambda(t) \, \partial_\lambda \mathcal{H}[\psi(t); \lambda(t)] = \beta \mathcal{W}[\psi; \lambda], \qquad (132)$$

and the action transforms as

$$\beta \mathcal{F}_\lambda(-t_0) + S_{\mathrm{cl}}[\psi, \hat{\psi}; \lambda] \overset{\mathcal{T}_\beta}{\mapsto} \beta \mathcal{F}_{\bar{\lambda}}(-t_0) + S_{\mathrm{cl}}[\psi, \hat{\psi}; \bar{\lambda}] + \Delta \mathcal{F}_{\bar{\lambda}} - \beta \mathcal{W}[\psi; \bar{\lambda}]. \qquad (133)$$

Transforming the left-hand side of $\langle 1 \rangle = 1$, we obtain the Jarzynski equality [42, 43]

$$\mathrm{e}^{\beta \Delta \mathcal{F}_\lambda} \langle \mathrm{e}^{-\beta \mathcal{W}[\psi; \lambda]} \rangle = 1. \qquad (134)$$

Starting from $P_\lambda(\mathcal{W}) = \langle \delta(\mathcal{W}[\psi; \lambda] - \mathcal{W}) \rangle$, the probability of the work to be $\mathcal{W}$ under the protocol $\lambda$, we obtain the Crooks fluctuation theorem on the statistics of the work [44–47],

$$P_\lambda(\mathcal{W}) = \mathrm{e}^{\beta(\mathcal{W} - \Delta \mathcal{F}_\lambda)} P_{\bar{\lambda}}(-\mathcal{W}). \qquad (135)$$

Compare the expressions in Eqs. (134) and (135) with their quantum versions derived in Eqs. (101) and (104), respectively.

## 6 Conclusions

In this manuscript, we contributed to a program that started half a century ago in the context of the stochastic dynamics of classical systems coupled to a thermal environment: we managed to identify the symmetry corresponding to *quantum* equilibrium dynamics within a Keldysh-like path-integral formalism. It is a universal (model-independent) discrete symmetry of the action which encodes the dynamics. It combines a time reversal and a KMS transformation of the fields. At the level of observables, the corresponding Ward-Takahashi identities provided a new derivation of the well-known quantum fluctuation-dissipation theorem in terms of a path-integral formulation.

This progress was made possible by our extension of the Keldysh field theory to arbitrary re-parametrization of time in the complex plane.

More importantly, we understood how non-equilibrium conditions, such as a time-dependent potential, break this symmetry and we interpreted the symmetry-breaking terms that are generated in terms of production of irreversibility. This allowed us to derive what is, we believe, the first proof of the quantum fluctuation theorems within a path-integral formulation, and to generalize the approach to derive new relations between multi-point correlation functions, a result not only physically but also technically important for quantum field theory. As a by-product, we identified the measure of the irreversibility $\mathcal{S}^{\mathrm{irr}}[\psi^+, \psi^-; \lambda]$ generated at the level of a single non-equilibrium quantum trajectory, thus solving a longstanding problem and bringing the field of stochastic thermodynamics to the realm of quantum physics.

The reformulation of equilibrium and non-equilibrium dynamics in terms of symmetry and symmetry breaking promises to be a powerful approach to analyze the non-equilibrium properties of quantum many-body systems. Indeed, these systems are typically intractable either analytically or numerically, and light-weight symmetry-based arguments could unlock some of the hard problems, especially in the context of thermalization.

## Acknowledgements

It is a pleasure to thank Denis Bernard and Jorge Kurchan for many stimulating discussions. LFC is a member of the Institut Universitaire de France.

## A  General idea in the operator formalism

To better illustrate the idea underlying the construction in Sect. 2, we express it here in a more concise manner within the operator formalism. Let us consider a quantum system described by the ket $|\psi(t)\rangle$ and a time-evolution governed by a time-dependent Hamiltonian $H(t)$. Instead of working in the original frame, we apply the following time-dependent U(1) rotation:

$$|\psi(t)\rangle \mapsto U(t)|\psi(t)\rangle \text{ with } U(t) = e^{i\theta(t)H(t)}, \tag{136}$$

where $\theta(t)$ is a generic real-valued function of time. Accordingly, the Hamiltonian transforms as

$$H(t) \mapsto H_\theta(t) \equiv H(t) + \int_0^1 dx\, e^{-xi\theta(t)H(t)}\, \frac{\partial}{\partial t}\big(\theta(t)H(t)\big)\, e^{xi\theta(t)H(t)}. \tag{137}$$

In the time-dependent frame, the Schrödinger equation naturally reads

$$i\partial_t|\psi(t)\rangle = H_\theta(t)|\psi(t)\rangle. \tag{138}$$

Note that reverting back to the original basis, *i.e.* $|\psi(t)\rangle \mapsto e^{-i\theta(t)H(t)}|\psi(t)\rangle$, corresponds to transforming

$$H_\theta(t) \mapsto e^{i\theta(t)\mathrm{Ad}_{H(t)}}H_\theta(t) - \int_0^1 dx\, e^{xi\theta(t)\mathrm{Ad}_{H(t)}}\, \frac{\partial}{\partial t}\big(\theta(t)H(t)\big) = H(t), \tag{139}$$

as expected.

In the special case of a constant Hamiltonian, $H$, we simply get

$$H_\theta(t) = [1 + \partial_t\theta(t)]H. \tag{140}$$

By re-parametrizing time as $t \mapsto \tau(t) \equiv t + \theta(t)$ (assuming the map between $t$ and $\tau$ is invertible) and by relabeling

$$|\psi(\tau)\rangle \equiv |\psi(t(\tau))\rangle, \tag{141}$$

the Schrödinger equation in the time-dependent frame can now be cast as a time-independent equation

$$i\partial_\tau|\psi(\tau)\rangle = H|\psi(\tau)\rangle. \tag{142}$$

This illustrates the covariance with respect to generic time re-parametrization.

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
