# Peer review of "(Non) equilibrium dynamics: a (broken) symmetry of the Keldysh generating functional"

_SciPost Physics, doi:SciPost Phys. 4, 008 (2018)_

## Round 2 · Referee Report · Anonymous (Referee 1) · 2017-7-30

Strengths

1- very clear and precise derivations; 2- effort in making a technical, hard topic understandable to the readers.

Weaknesses

1- lacking of a concrete (even simple) application to a model of interest.

Report

This work extends previous lore on symmetry properties exhibited by the Keldysh action for systems at thermodynamic equilibrium. The topic has a long tradition, as also pointed out by the authors, and its main practical importance stands in the fact that non-equilibrium conditions break this symmetry and novel symmetry-breaking terms appearing in the action can help in enlarging physical insight. This is achieved by the authors, extending to time-dependent quantum systems pre-existent expertise on the subject.

A nice feature of the work is its readability by field-theoretical experts that are not familiar with Keldysh formalism: the authors have done some effort in making their derivations clear step-by-step. This, however, has the downside that the community of quantum thermodynamics (which should be one of the main beneficiary of their findings) might have an hard time in understanding the core results. In view of this, and of the fact that ultimately such technical works should always present a spectrum of interesting applications, I would recommend the authors to go after the following suggestions:

1) a summary on physical findings and outreach of applications should be substantially enlarged in the introduction;

2) they should find and add to the manuscript a concrete application of Eqs. (92-93-94), even a simple one (for example: one body, quantum, coupled to a thermal bath and to a time-dependent external field), where the predictions of their main results can become of relevance for the debate (in the quantum thermodynamics community) concerning a proper, suitable definition of work and produced entropy in non-equilibrium settings.

This would help their work and their results to be easier spread and employed by the communities of interest.

Finally, as a speculative point, one might wonder whether the equilibrium symmetry can be spontaneously broken in some particular circumstances. This comes naturally to the mind after reading their work, since they realise an explicit breaking by adding an external time-dependent drive. Do the authors have some insight in this direction? Would they like to add some comments in the conclusions about it?

Minor observations: - in (15) the ‘;’ should be a typo; - what do the authors mean by ‘thermal rotation’ in section 2.3? - in 3.2 they mention the ‘thermo-field’ formalism, which is always contrasted with the Keldysh one. Since the authors show an understanding about the differences between the two, they have a chance to clarify for the community this point, which to my understanding has not been yet well inspected; for instance, they can contextualise it, when they compare their main finding in the two formalisms (between Eqs. 42 and 43).

Requested changes

1- extend summary on physical contents; 2- provide a workable, interesting example.

---

## Round 2 · Referee Report · Anonymous (Referee 2) · 2017-10-3

Strengths

1-Innovative formalism 2-Detailed discussion of the latter 3- Interesting results on the characterisation of irreversibility

Weaknesses

1- Hard to read 2- Too technical for non-experts.

Report

The manuscript by C. Aron et al. presents a generalization of the standard Schwinger-Keldysh field theoretic formalism which involves a re-parametrization of Keldysh contours (including the equilibrium branches) to generic contours in the complex plane. While similar analytic continuations have been historically used for example to derive various relations among Green’s functions and their convolutions, this idea is here exploited to its full power to derive universal symmetry relations satisfied by equilibrium theories and how taking the system out of equilibrium breaks them.

I think that overall the content of the paper and its results are very interesting and the paper should be published with minor revisions.

In particular, the paper is extremely technical and its style makes it at the moment hardly readable to non-experts. It took me some time to navigate through it and digest its various parts. In particular the main physical result, which is in my opinion contained in Eq.(88)-(91) (and the following fluctuation theorems), is very interesting. It is however only briefly announced in Sec.1.3 (referring to Eq.(102) though a lot of physics is contained much before that) as a “general expression for entropy production”, and later on as a “quantum generalisation of dissipated work” ( introduction of Sec.4), while it is very hard to see either one or the other in the concrete expressions, apart of course for the “slow” limit Eq.(95). The physical meaning of $\Sigma$ is still obscure the connection with other quantities and their fluctuation theorems appears unclear (e.g. Physical Review E 75, 050102(R) (2007) and generalisations).

In order to make the paper more readable, I would suggest to significantly revise the section 1.3 about “Main Results” which is now just an extended summary of the paper focusing for the greatest part on the formalism. I would suggest to rather put significant stress on the section on non equilibrium dynamics and on $S^{irr}$ or $\Sigma$ discussing its context, why it is important (including references), clarifying its physical meaning or at least its meaning in the various limits in some detail.

Requested changes

1- Revision of Sec.1.3 as detailed in my report

---

## Round 3 · Referee Report · Anonymous · 2017-12-21

Report

The authors have seriously taken into consideration my advice contained in the previous report. I am glad to assess that now the paper is an interesting and complete piece of science, with both technical and practical aspects of interest for the community working in non-equilibrium many body physics.

---

## Round 3 · Author Response

We are much grateful to both Referee 1 and Referee 2 for the time they spent writing a thorough, positive, review of our work. The appreciations of both Referees being quite similar, we answer them in a single reply.

While both Referees acknowledged the significance and the originality of our work, they also pointed out that, given the rather formal nature of our approach, additional efforts could be made to better present the physical findings. We fully agree with the Referees, and we thank them for their suggestions in that respect, which we followed (see the List of changes).

We hope that the referees will find the revised version of our manuscript suitable for publication in SciPost without further delay.

Camille Aron
Giulio Biroli
Leticia F. Cugliandolo

---

## Round 3 · List of Changes

- We revised the introductory Section 1.1 "Motivations" to better appeal to the community of quantum thermodynamics by articulating one of our main findings, namely the identification of a quantum version of the irreversible entropy production rate, in the context of what has already been achieved in the classical realm (stochastic thermodynamics), and of the questions that are still debated in the quantum realm.

- We revised the introductory Section 1.3 "Main results", paragraph "Out-of-equilibrium dynamics", by announcing a later discussion on a concrete and simple model where our findings can be worked out and discussed explicitly, numerically or even experimentally.

- We added the corresponding discussion in Sect. 4.2 "Symmetry breaking term", where we consider a two-level system (TLS) which is driven through an avoided crossing at different rates. Despite its simplicity, the time-dependent driven TLS has two virtues:
(i) it belongs to a different `class' of systems than the bosonic and fermionic fields that are treated in the rest of the manuscript. The TLS is an SU(2) symmetric object with a more cumbersome path-integral representation, however the quantum thermodynamic operators that were derived using bosonic and fermionic fields are readily applicable to any quantum mechanical system,
(ii) the excitations created when ramping across the avoided crossing at a finite rate are of purely quantum mechanical origin (contrary to the case of, say, a driven quantum harmonic oscillator which comes with a 'classical' intuition/phenomenology). While the evolution is purely unitary, it is clear that these tunnel excitations can be associated to the production of irreversibility in the system.
The novel quantity $\langle \Sigma \rangle$ is studied numerically in different regimes, and compared to the work $\langle \mathcal{W} \rangle$. This simple model is used again in Section 4.3 "Quantum fluctuation theorems", paragraph "Jarzynski equality", to illustrate our proposed definition of irreversibility production, $\mathcal{S}_{irr}$, given in Eqs. (100) and (104).

You are currently on this page

Resubmission 1705.10800v3 on 21 December 2017

---

## Editorial Decision

published